# INFINIBENCH: A COMPREHENSIVE BENCHMARK FOR LARGE MULTIMODAL MODELS IN VERY LONG VIDEO UNDERSTANDING

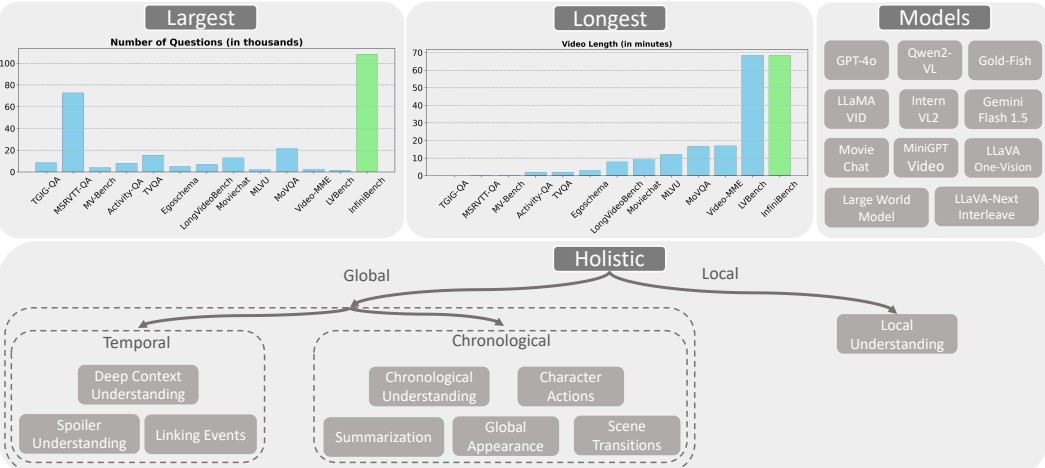

Figure 1: An overview of InfiniBench characteristics: 1) The largest as it contains more than 100K questions. 2) The longest as it consists of movies that exceed 1 hr duration. 3) Holistic by covering nine diverse skills and 11 models including commercial and open-source ones.

## ABSTRACT

Understanding long videos, ranging from tens of minutes to several hours, presents unique challenges in video comprehension. Despite the increasing importance of long-form video content, existing benchmarks primarily focus on shorter clips. To address this gap, we introduce InfiniBench a comprehensive benchmark for very long video understanding which presents *1)very long* video duration, averaging 52.59 minutes per video; *2) The largest* number of question-answer pairs, 108.2K; *3) Diversity* in questions that examine nine different skills and include both multiple-choice questions and open-ended questions; *4) Memory questions*, such as Global Appearance that require remembering and tracking the visual aspects through the video. Using InfiniBench, we comprehensively evaluate existing Large Multi-Modality Models (LMMs) on each skill, including the commercial models such as GPT-4o and Gemini 1.5 Flash and the recent open-source models. The evaluation shows significant challenges in our benchmark. Our findings reveal that even leading AI models like GPT-4o and Gemini 1.5 Flash face challenges in achieving high performance in long video understanding, with average accuracies of just 56.01% and 43.32%, and average scores of 3.25 and 2.79 out of 5, respectively. Qwen2-VL matches Gemini's performance in the MCQ skills but lags significantly in open-ended question tasks. We hope this benchmark will stimulate the LMMs community towards long video and human-level understanding. Our benchmark can be accessed at InfiniBench.

## 1 INTRODUCTION

Recent Large Language Models (LLMs) (Li et al., 2023a; Achiam et al., 2023; Touvron et al., 2023) have shown impressive progress in the Natural Language community. Inspired by the strong abilities

of LLMs, Large Multi-Modality Models (MLMMs) (Ataallah et al., 2024a; Zhu et al., 2023; Zhang et al., 2023a; Chen et al., 2023; Lin et al., 2023; Liu et al., 2023b; Maaz et al., 2023; Bai et al., 2023) which equip the LLMs with visual processors have been developed to solve cross-modality tasks such as image understanding and short video understanding. While current large multi-modality models show some progress in video understanding, their abilities remain unclear for very long-form video understanding.

Long-form video understanding (Song et al., 2023; Regneri et al., 2013; Rohrbach et al., 2014; Awad et al., 2017; 2018; 2020) not only challenges these models by increasing the number of images but also contains more comprehensive information, making it a boundary-pushing task toward human-level intelligence. For example, humans can link multiple events at different times and answer questions requiring a deep understanding of events or characters in a long video. Multi-modal models can address these questions, requiring long-range temporal-spatial reasoning and strong vision-language alignment abilities, potentially serving a wider range of AI applications.While the necessity of a long-video understanding benchmark is evident, the current recent benchmarks are up to 17 minutes, however LVBench (Wang et al., 2024) introduces a 1-hour-long benchmark with 1,549 QA pairs.it only supports visual mode, whereas our benchmark examines the more challenging combined subtitle + visual understanding capability. Additionally, the scale of our benchmark is 70× larger than LVBench.. To fill this gap in comprehensive long video understanding, we propose InfiniBench, a comprehensive benchmark for very long-form video understanding. As shown in Tab 1, InfiniBench is currently the video benchmark that has both the longest length (52.59 minutes) and the largest number of question-answer (QA) pairs (108.2K). The video sources are movies and daily TV shows, and the questions are designed based on multiple sources including video frames, video scripts, and video summaries. As shown in Figure 1 , the QA pairs consist of nine carefully designed types of questions, which mainly focus on human-centric aspects, including Summarization, Global Appearance, Scene Transitions, Sequence of Actions by Each Character, Temporal Questions, Linking Events, Deep Context Understanding, Movie Spoiler Questions, Local Visual and contextual Questions. The questions include multiple-choice questions (MCQs) and open-ended questions. We report the accuracy for MCQs and the GPT-4o rating score for open-ended questions. The annotation process is mainly done by an automatic pipeline using GPT-4, which includes proposing questions and generating answers. To prevent hallucinations and gather sufficient information for generating QA pairs, we used various sources of information including video frames which are used in the global appearance skill, video transcripts, and video summaries.

Based on InfiniBench, we evaluate the current state-of-the-art MLLMs capable of handling very long videos, including the open-source models Movie-Chat(Song et al., 2023), Llama-Vid(Li et al., 2023b), Large World Model(Liu et al., 2024),Qwen2-VL(Bai et al., 2023) and the commercial models such as GPT-4o and Gemini 1.5 Flash.

We summarize the key experimental findings here: (1) All existing models struggle with InfiniBench, showing the unique challenges of our benchmark. (2) Experiments show that GPT-4o outperform all open-source models on each skill with a large gap despite it struggles with average accuracies of just 56.01% and, and average scores of 3.25 out of 5.(3)Only Qwen2-VL surpasses Gemini in the MCQ skills but struggles in the open-ended.

(4) All models showed better performance in local skills compared to global skills. (5)The most difficult skill in MCQ is character actions, while for open-ended questions, it is movies spoiler questions that designed for human-centric video understanding.

By introducing this comprehensive InfiniBench, we hope to:

- Help bridge the gap of lacking a large-scale long-form video understanding benchmark.
- Boost the development of current open-source LMMs.
- Push MLMMs towards human-centric and human-level long video understanding.

## 2 RELATED WORK

**Short video benchmarks**     The previous short and long videos benchmarks are listed in Table 1. Short video benchmarks have been extensively studied in (Maaz et al., 2024; Jang et al., 2019; Xu et al., 2017; Lei et al., 2019; Miech et al., 2019). Although HowTo100M(Miech et al., 2019) is

| Category | Models | # Questions | # Videos | Video Duration | Global Questions | Questions Type | | QA Source | | | Annotations | |
|---|---|---|---|---|---|---|---|---|---|---|---|---|
| | | | | | | MCQ | Open | Video | Transcript | Summary | Auto | Human |
| Short | TGIF-QA | 8.5 K | **9575** | 0.05 | ✓ | ✗ | ✓ | ✓ | ✗ | ✗ | ✓ | ✓ |
| | MSRVTT-QA | 72.8 K | 2990 | 0.25 | ✗ | ✗ | ✓ | ✓ | ✗ | ✗ | ✓ | ✗ |
| | MV-Bench | 4.0 K | 3641 | 0.27 | ✓ | ✓ | ✗ | ✓ | ✗ | ✗ | ✓ | ✗ |
| Long | Activity-QA | 8.0 K | 800 | 1.85 | ✗ | ✗ | ✓ | ✓ | ✗ | ✗ | ✗ | ✓ |
| | TVQA | 15.2 K | 2179 | 1.86 | ✗ | ✓ | ✗ | ✓ | ✗ | ✗ | ✗ | ✓ |
| | Egoschema | 5.0 K | 5063 | 3.00 | ✓ | ✓ | ✗ | ✓ | ✗ | ✗ | ✗ | ✓ |
| | LongVideoBench | 6.7 K | 3763 | 7.88 | ✗ | ✓ | ✗ | ✓ | ✗ | ✗ | ✗ | ✓ |
| | Moviechat | 13.0 K | 1000 | 9.40 | ✓ | ✗ | ✓ | ✓ | ✗ | ✗ | ✗ | ✓ |
| | MLVU | 2.6 K | 757 | 12.00 | ✓ | ✓ | ✓ | ✓ | ✗ | ✗ | ✓ | ✓ |
| | MoVQA | 21.9 K | 100 | 16.53 | ✓ | ✓ | ✗ | ✓ | ✗ | ✗ | ✗ | ✓ |
| | Video-MME | 2.7 K | 900 | 16.97 | ✓ | ✓ | ✗ | ✓ | ✗ | ✗ | ✗ | ✓ |
| Very Long | LVBench | 1.6 K | 103 | **68.35** | ✓ | ✓ | ✗ | ✓ | ✗ | ✗ | ✗ | ✓ |
| | **InfiniBench (Ours)** | **108.2 K** | 1219 | 52.59 | ✓ | ✓ | ✓ | ✓ | ✓ | ✓ | ✓ | ✓ |

Table 1: Comparison between InfiniBench and existing video understanding benchmarks (TGIF-QA (Jang et al., 2017), MSRVTT-QA (Xu et al., 2017), MV-Bench (Li et al., 2024b), Activity-QA (Yu et al., 2019), TVQA (Lei et al., 2019), Egoschema (Mangalam et al., 2023), LongVideoBench (Song et al., 2023), Moviechat (Song et al., 2023), MLVU (Zhou et al., 2024), MoVQA(Zhang et al., 2023b), Video-MME(Fu et al., 2024), and LVBench(Wang et al., 2024)). InfiniBench has the largest QA pairs, the most videos, and the longest average duration. (Note: Global Q stands for whether any challenging questions are designed to explain the whole video. VS is the video's script, and VSum is the summary of the video.)

a valuable and diverse repository of daily life actions with an extensive collection of 136 million videos, it suffers from weak labeling due to its reliance on narrative subtitles and its average video duration is 3.6 second. MSRVTT-QA (Xu et al., 2017) has a large number of questions, but it does not support global questions, and the annotations are automatically generated without human verification. TGIF-QA (Jang et al., 2017) and MV-Bench (Li et al., 2024b) are short video benchmarks that support global questions, but their scale is limited. Activity-QA (Yu et al., 2019), TVQA (Lei et al., 2019), do not support global and have only local questions. Egoschema (Mangalam et al., 2023) is a human-annotated Long-form Video understanding Benchmark and the video length are only three-minute-long.

**Long video benchmarks** MovieChat-1K (Song et al., 2023) is a benchmark dataset derived from movies, featuring 1,000 video clips spanning various genres with an average duration of 9.4 minutes. It includes 14,000 annotations designed to support diverse visual narratives and question-answering tasks.

Recently, researchers continue to push the boundaries of video lengths in long video understanding benchmarks (Zhou et al., 2024; Zhang et al., 2023b; Fu et al., 2024; Wang et al., 2024). MLVU (Zhou et al., 2024) includes diverse types of videos and tasks . MoVQA (Zhang et al., 2023b)is a benchmark fully sourced from movies, designed with multi-temporal-level questions. Video-MME (Fu et al., 2024)is a high-quality long video benchmark annotated by expert annotators, featuring various video lengths. The video durations for these benchmarks have a maxmium average duration of approximately 17 minutes. Compared to these works, our dataset offers several advantages: (1) The video lengths in our dataset approach 1 hour. (2) Our dataset is significantly larger in scale. (3) It supports both multiple-choice questions (MCQ) and open-ended evaluations. (4) It includes the video script and a summary of the video as additional resources for question answering. We also acknowledge a contemporary effort, LVBench (Wang et al., 2024), which presents a benchmark comprising 1,549 question-answer (QA) pairs derived from diverse video sources with average duraion of one hour.In comparison, Infinibench significantly surpasses LVBench in the scale of QA pairs by approximately 70 time.

**Long Video Models.** Google Gemini-Flash 1.5 model (Gemini, 2024) is currently the only available native commercial model capable of processing extremely long videos, boasting an unprecedented context window of 1 million tokens.This extensive context window allows Gemini-Flash 1.5 to effectively handle both video frames and subtitles simultaneously. GPT-4o (OpenAI, 2024) can also handle video input by processing up to 250 frames along with the accompanying subtitles, ensuring that the combined data fits within the model's maximum context window of 128K tokens. In contrast to the commercial solutions, there is some recent open source models that can process long videos such as Qwen2-VL (Bai et al., 2023) :Qwen2-VL is a recently developed multimodal model designed to process images at multiple resolutions. It is also capable of handling videos with a maximum of 768 frames, making it inherently suited for long video processing. The model

leverages Multimodal Rotary Position Embedding (M-RoPE), an extension of the one-dimensional RoPE, to effectively encode positional information for both images and videos, enhancing its ability to understand spatial and temporal relationships. Goldfish (Ataallah et al., 2024b)is an efficient retrieval framework designed to handle arbitrarily long videos by segmenting them into multiple clips and retrieving the top-k clips most relevant to the query. LLama-vid (Li et al., 2023b) is a recent open-source model that comprehends long videos due to its excellent efficiency in representing each frame using only two tokens. The Large World Model (LWM) (Liu et al., 2024) is another open-source model capable of processing millions of tokens using the innovative ring attention mechanism (Liu et al., 2023a). Consequently, Moviechat (Song et al., 2023) processes long videos but without subtitles and operates in two modes: global and breakpoint. The global mode exclusively utilizes long-term memory, and the breakpoint mode additionally incorporates the current short-term memory as part of the video representation. The breakpoint mode allows for understanding the video at a specific moment in time.

## 3   INFINIBENCH

In this section, we first dissect the data collection pipeline (Sec 3.1) then the skills definition (Sec 3.2), and finally, the benchmark statistics (Sec 3.3).

### 3.1   DATA COLLECTION

We utilized two sources to obtain very long videos: Movies and TV shows, for Movies, we employed the MovieNet dataset (Huang et al., 2020). However, no dataset is available for complete TV shows, as TVQA(Lei et al., 2019) provides only short clips,Inspired by Goldfish (Ataallah et al., 2024b) trick, we transformed the TVQA dataset from a collection of short clips into a long video dataset by gathering and sequencing the clips corresponding to each episode, thereby reconstructing the full episode frames. We obtained 924 full-length episodes from six different TV shows through this modification. Consequently, MovieNet dataset(Huang et al., 2020), has only 296 movies which had shots aligned with subtitles. Therefore, only these movies are included; we excluded the rest from our benchmark. In addition, we relied on two extra data sources: the video summaries and transcripts. For the TVQA dataset (Lei et al., 2019), the summaries from IMDB and the transcripts were scraped for the 924 episodes. For the filtered MovieNet movies (296)(Huang et al., 2020), we obtained transcripts from the MovieNet annotations. However, since MovieNet (Huang et al., 2020) annotations do not include complete movie summaries, the missing summaries are scrapped from IMDB to obtain comprehensive movie summaries and transcripts for all filtered movies.

For spoiler skill, out of 296 movies in the MovieNet (Huang et al., 2020) dataset, we identified 147 movies with associated spoiler questions available on IMDb, totaling 806 questions. These questions were meticulously collected and integrated into our benchmark dataset. Consequently, we directly adopted TVQA questions for the local skills by aggregating questions corresponding to clips from the same episode, ensuring multiple questions per episode. Notably, these questions in TVQA (Lei et al., 2019) exhibit a dual property encompassing visual and contextual dimensions. It's pertinent to mention that these questions are exclusive to the TVQA dataset and have hitherto remained unutilized for long video benchmark solely for analyzing short clips.

### 3.2   SKILLS

To create a robust benchmark for long video understanding, the questions should cover both local and global events throughout the video. As shown in Figure 1, the global questions can involve temporal or chronological reasoning, requiring a deeper understanding of the sequence and progression of events to provide accurate answers. This dual focus ensures a comprehensive evaluation of a model's ability to grasp both detailed and holistic aspects of video content.

Additionally, figures of skills are presented in the Supplementary. E

#### 3.2.1   GLOBAL CHRONOLOGICAL SKILLS

**Global Appearance.**   In this skill, we focused on generating questions that require continuous visual understanding, which cannot be answered from short video segments but necessitate watching

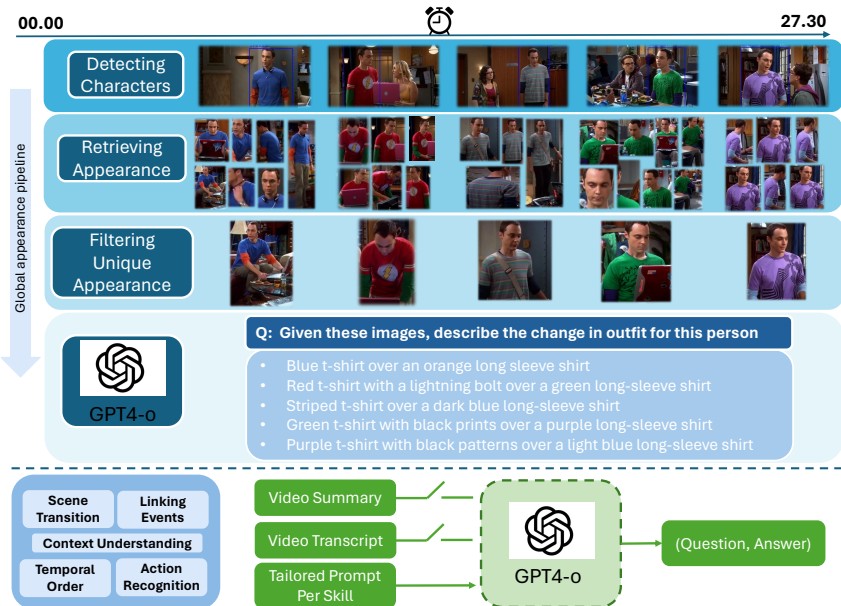

Figure 2: Full annotation pipeline for InfiniBench skill set. The upper section depicts the global appearance pipeline, while the lower section illustrates the question generation using GPT-4o. The gates for video summary and video transcript indicate that some skills utilize only the summary, others use only the transcript, and some use both.

the entire video. We selected changes in outfits as the basis for these continuous vision questions. To create this type of question, we developed the global appearance pipeline, as shown in Figure 2. The TVQA+ (Lei et al., 2020) dataset was used, providing bounding boxes for each character in one of the six TV shows in TVQA (Lei et al., 2019), specifically *The Big Bang Theory*. Images were cropped using these bounding boxes, and all images of each character in the episode were collected. Manual filtering was performed to select each character's best and most unique outfits. GPT-4o described the outfit for each unique image and generated a sequence list of the outfits For evaluation, multiple-choice questions were formulated by altering the sequence of outfits. For example: "Choose the correct option for the following question: In what order does Leonard change outfits in this episode?" The correct option is (a) a red T-shirt under a beige jacket with a green hood, a white t-shirt with a green print under a grey jacket and black vest, or a white dress shirt with a patterned tie under a brown blazer. Other options present the outfits in the incorrect order. In special cases where a character's outfit does not change throughout the episode, distractor options with incorrect outfits were added as alternative choices.

**Scene Transitions.** Scene transition skills necessitate continuous visual comprehension and cannot be adequately addressed using short video segments; they require viewing the entire video. To assess this skill, questions concerning transitions between scenes were generated. It was observed that the locations of each scene are mentioned in the transcript. Utilizing GPT-4o by inputting the transcript of the TV shows as in Figure 2. We extracted these locations and created a list in the correct sequence. Then, for evaluation, we follow a template-based approach to collect multiple-choice questions to assess the correct sequence of these scene transitions.

**Character Actions** This skill involves generating questions about each character's actions, encompassing contextual and visual actions, which can often be identified in the transcript where scene actions are described. For example, "Rachel serving coffee to her friends in Central Park." To create these questions, we utilized GPT-4o by inputting both the video summary and the transcript as in Figure 2. This approach ensures that the questions accurately reflect the sequence of actions depicted in the video. To evaluate this skill, we formulated multiple-choice questions regarding the correct order of actions performed by each character. These questions were generated for both the Long TVQA and MovieNet datasets.

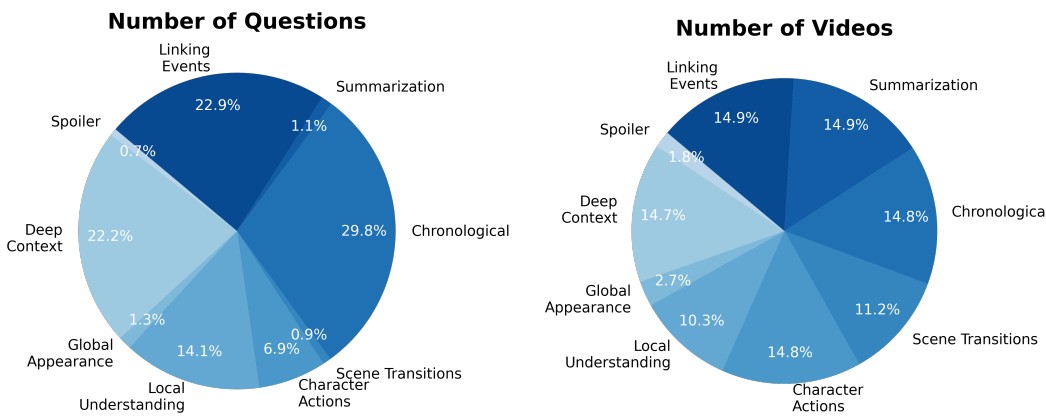

Figure 3: Left) Number of questions distribution for each skill set. Right) Number of videos for each skill.

**Chronological Understanding**    This skill assesses the temporal understanding of long videos by generating questions about the correct sequence of events in movies or TV series, and these events cover both visual and contextual events. We ask questions regarding which event occurred first or the correct order of adjacent events. For instance, "Is event A before event B?" or "What is the correct sequence of these events: event A, event B, or event C?" To generate these questions, we utilized GPT-4o by inputting the episode's transcript as in Figure 2. We used the transcript instead of the summary, as the correct order of events can only be accurately extracted from the detailed transcript. These questions are presented in a multiple-choice format and generated for both the Long TVQA we created and MovieNet (Huang et al., 2020) datasets.

**Summarization.**    Summarization is a critical skill for evaluating long sequence data, such as long text understanding in NLP, and is equally crucial for assessing long video comprehension. Our benchmark includes human-generated summaries for movies and TV shows sourced from IMDb. These summaries, created by humans, encapsulate visual and contextual events in the videos, making it a strong skill for evaluating a long video understanding.

### 3.2.2 GLOBAL TEMPORAL SKILLS

**Deep Context Understanding.**    For this skill, we aim to test the model's ability to answer hard and tricky questions requiring a deep understanding of the full video. We utilized GPT-4o to generate challenging and nuanced questions about the video. We did not restrict GPT-4o to a specific skill set, allowing the advanced AI model to generate questions autonomously. We provided GPT-4o with comprehensive information about the video, including the transcript and summary as in Figure 2, enabling it to create complex questions that require a profound understanding of the context and the main topic of the movie or the TV show. These open-ended questions were developed for the Long TVQA we created and MovieNet(Huang et al., 2020) datasets.

**Movies Spoiler Questions.**    Spoiler questions are inquiries that reveal critical plot points, twists, or specific details that could potentially spoil the experience for viewers who have not yet seen the movie. These questions are crucial for evaluating long videos because they delve into significant, often pivotal moments in the narrative, requiring a deep and comprehensive understanding of the entire storyline. These questions are important for long video evaluation for several reasons:

- *Comprehensive Understanding*: Answering spoiler questions necessitates a thorough comprehension of the entire video, as they often reference events from various points in the narrative. This ensures that the evaluator has engaged with the content meaningfully and sustainably.
- *Critical Thinking*: These questions require viewers to think critically about the plot and its developments, analyzing character actions and narrative resolutions.

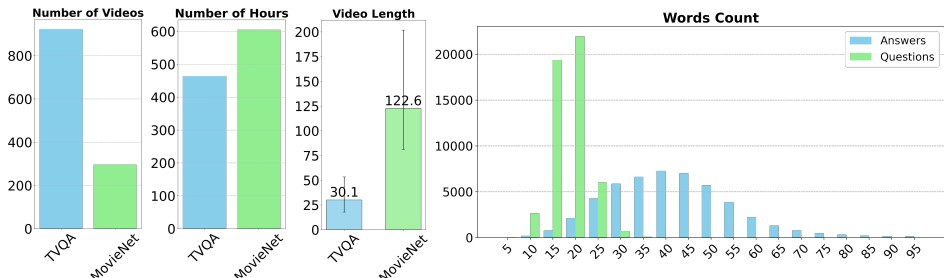

Figure 4: Data statistics. The first two figures On the left,reporting the number of videos and their length in hours from each data source: TVQA and MovieNet datasets. In the middle, we demonstrate the video duration of each video source. On the right, we show the histogram of the lengths of the questions and answers.

- *Detail Orientation*: Spoiler questions often focus on specific, detailed aspects of the plot, ensuring that the evaluator has paid close attention to the video.

**Linking Events.** This skill involves generating a set of questions that link multiple events together, such as events from the beginning of an episode that affect later events, to ensure the questions comprehensively cover the entire video. Examples of such questions include:

- What is the influence of event A on event B?
- How does event A lead to event B?
- What is the relationship between event A and event B?
- What is the impact of event A on event B?

We generated these questions by inputting the video summary into GPT-4o and instructing GPT-4o to create this type of question as in Figure 2. These open-ended questions were developed for the Long TVQA we created and MovieNet(Huang et al., 2020) datasets.

### 3.2.3 LOCAL UNDERSTANDING

**Local Understanding** The local questions in our benchmark are adapted from the TVQA dataset Lei et al. (2019) and are inspired by the Goldfish paper Ataallah et al. (2024b) that converted the evaluation from short video to long videos. These questions are multiple-choice and rely on human annotations. The local questions are specifically designed to assess the model's ability to localize and focus on specific segments within extended video content. Successfully answering these questions indicates the model's capacity to capture fine-grained details, making it analogous to a "needle-in-a-haystack" challenge, but tailored for video understanding.

### 3.3 BENCHMARK STATISTICS

The InfiniBench benchmark represents the largest dataset for long video question-answering, comprising 108.2K questions spanning nine distinct skill categories. Figure 3 (left) displays the distribution of questions across these skills, while Figure 3 (right) illustrates the distribution of videos associated with each skill.

Figure 4 (left) provides an overview of the number of videos and total hours of footage included from each data source in the benchmark.The center of the Figure 4 highlights the variation in video durations across the benchmark's video sources, such as TVQA (Lei et al., 2019) and MovieNet (Huang et al., 2020).Notably, the benchmark includes videos with durations ranging up to 201 minutes (3.35 hours), underscoring its focus on long video understanding. ,On the right, a histogram shows the distribution of word counts for the questions and answers.

Also, human verification for InfiniBench is presented in Supplementary with overall correctness og 95.8%. A.

| Models | Frame Rate | Subtitles | Global Appearance | Scene Transitions | Character Actions | Chronological Understanding | Local Understanding | Summar- ization | Deep Context Understanding | Spoiler Understanding | Linking Events | Avg. Acc. | Avg. Score |
|---|---|---|---|---|---|---|---|---|---|---|---|---|---|
| Random Accuracy | – | – | 16.68 | 16.66 | 16.14 | 41.51 | 20.00 | – | – | – | – | 22.20 | – |
| GPT-4o | 250 FPV | ✓ | **45.39** | **47.93** | 36.07 | **68.85** | **81.75** | **3.49** | **3.39** | **2.67** | **3.45** | **56.00** | **3.25** |
| Qwen2VL | 250 FPV | ✓ | 38.39 | 37.54 | **36.86** | 50.85 | 59.98 | 0.67 | 2.07 | 1.41 | 2.76 | 44.72 | 1.73 |
| Gemini Flash 1.5 | - | ✓ | 31.80 | 31.63 | 37.82 | 56.41 | 58.95 | 3.24 | 2.55 | 2.05 | 3.33 | 43.32 | 2.79 |
| LLaVA-OneVision | 128 FPV | ✓ | 38.60 | 25.02 | 24.83 | 45.91 | 48.54 | 0.49 | 1.78 | 1.30 | 2.51 | 36.58 | 1.52 |
| InternVL2 | 128 FPV | ✓ | 29.26 | 21.98 | 25.00 | 44.63 | 42.62 | 0.69 | 1.68 | 1.25 | 2.47 | 32.70 | 1.52 |
| LLaMA-VID | 1 FPS | ✓ | 17.37 | 17.06 | 18.25 | 41.74 | 23.73 | 1.58 | 2.00 | 1.49 | 2.40 | 23.63 | 1.87 |
| Goldfish | 0.5 FPS | ✓ | 10.30 | 2.82 | 20.87 | 40.14 | 40.66 | 0.77 | 2.36 | 1.85 | 3.01 | 22.96 | 2.00 |
| Large World Model | 8 FPV | ✗ | 9.20 | 3.30 | 8.46 | 38.64 | 18.70 | 0.02 | 0.90 | 0.60 | 0.89 | 15.66 | 0.60 |
| MovieChat | L/8 FPV | ✗ | 8.10 | 7.60 | 4.67 | 38.20 | 18.27 | 0.14 | 0.62 | 0.41 | 1.00 | 15.37 | 0.54 |
| LLaVA-Next Interleave | 8 FPV | ✗ | 1.71 | 0.32 | 0.44 | 41.90 | 18.75 | 0.28 | 1.33 | 0.92 | 1.87 | 12.62 | 1.10 |
| MiniGPT4-video | 45 FPV | ✓ | 2.33 | 1.09 | 2.36 | 39.86 | 15.15 | 0.05 | 0.54 | 0.75 | 0.89 | 12.16 | 0.56 |

Table 2: InfiniBench leader-board over the nine skills. FPV is the Frame rate Per Video. FPS is the Frame rate Per Second.

## 4 EXPERIMENTS

### 4.1 EVALUATION METRICS

We employed distinct evaluation metrics appropriate for the two questions types: open-ended and multiple-choice (MCQs). For MCQs, accuracy was the chosen metric, while for open-ended questions, we utilized a scoring system based on GPT-4-mini, ranging from 0 to 5. For MCQ, GPT-4-mini is used to match the predicted answer with one of the options or to match with the "I don't know option", that indicates there is no match or hallucination. See Sec. Evaluation Details in the supplementary D for more details. For open-ended questions, GPT-4-mini evaluated the LLMs' predictions based on multiple criteria: correctness, meaningfulness, proximity to the expected answer, presence of hallucinations, and completeness. Based on these criteria, GPT-4-mini generates a score ranging from 0 to 5, reflecting the overall quality of the response.

### 4.2 EVALUATION MODELS SETTING

In our evaluation, we evaluated two commercial models and nine open-source models.

**GPT-4o.** GPT-4o cannot natively process .mp4 video files. To work within its limitations, we sampled the maximum of 250 frames from the video, followed by the subtitle text and the question. **Gemini-Flash 1.5.** The Gemini-Flash 1.5 model, developed by Google (Gemini, 2024), Gemini recently gained the capability to process .mp4 files, but since our benchmark videos lack audio, we provided Gemini with the video file, followed by the subtitle file and the accompanying question. **Qwen2-VL** supports processing up to 768 frames; however, this exceeds the memory capacity of a single A100 GPU (80 GB) and requires the use of multiple GPUs. To ensure a fair comparison with GPT-4o, we limit the maximum number of frames to 250. Subtitles are provided as additional context, concatenated with the question, to enhance input comprehension. **InternVL2**: For the configuration of InternVL2, the default number of frames was increased to the maximum supported by the A100 GPU (80 GB), resulting in a limit of 128 frames. While the model can accommodate more than 128 frames, this often leads to significant hallucinations in its outputs. Performance was evaluated at different frame rates, specifically 16 and 128 frames. Subtitles were incorporated as additional context alongside the input question to enhance the model's understanding. **LLaVA-OneVision**: This model is capable of processing both videos and images. We investigated the maximum number of frames that can be processed using an A100 GPU (80 GB) without inducing hallucinations and evaluated the model's performance at frame rates of 16 and 128. Subtitles were included as additional context, provided alongside the input question to enhance comprehension. **Goldfish.** we used the default model setting with number of retrieved videos k=3 and process the video in 0.5 fps as each small clip is 90 sec and the model sample 45 from them which means 0.5 fps. **LLama-vid.** The LLaMA-VID model (Li et al., 2023b) accepts both video frames and subtitles. For our evaluation of the movies, we utilized our dataset with one frame per second, accompanied by aligned subtitle shots. The model was evaluated using the default settings without any modifications to the inference parameters. **Large World Model (LWM).** LWM is efficiently optimized for execution on Google TPUs and has another version for GPUs. Our evaluation is done using (NVIDIA A100), which allows for processing a maximum of 8 frames per video. While this setup does not represent the optimal configuration for LWM, it was the most feasible setting. LWM can accept only the video frames without the subtitles. **Moviechat.** The MovieChat model (Song et al., 2023) processes video frames without subtitles and operates in global

and breakpoint modes. Our evaluation focused on the global mode, utilizing the default inference settings without any modifications, where this setting allows to input the video length/8 frames. **LLaVA-NeXT-Interleave.** LLaVA-NeXT-Interleave can process only 8 frames per video, we also used the default setting without any changes. **MiniGPT4-video.** We evaluated MiniGPT4-video with the Llama 2 version that capable of handling 45 frames per video.we also used the default setting without any changes.

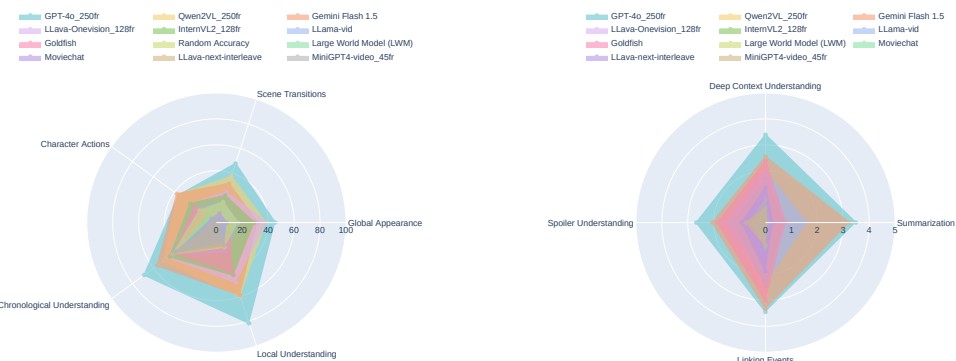

Figure 5: InfiniBench full leaderboard: Left – Comparison of performance across (MCQ) skills. Right – Evaluation of performance on open-ended question skills.

## 4.3 RESULTS

In this section, we evaluate the state-of-the-art (SOTA) open-source and commercial models for long video understanding. First, we present the overall performance averaged across all nine skills. Then, we delve into the specific skill performance detailed in Table 2, **Overall performance.** The overall performance of different models on InfiniBench is shown in Table 2 (j). Three findings can be observed: (1) All models' is relatively lower than other benchmarks (e.g., Movie-chat, MLVU,VideoMME benchmarks). This could be interpreted by the challenging nature of our skills that require deep, long-term understanding. To further verify this point, we test our benchmark on the most recent short-video models, e.g., MiniGPT4-video and LLaVA-NeXT-Interleave. We argue that short-video models should suffer if the benchmark truly assesses long-video understanding capabilities. In other words, the limited context captured by the short video models should not be enough to answer long reasoning queries. As shown in the table 2, MiniGPT4-video and LLaVA-NeXT-Interleave match lower than the random performance, which shows the effectiveness of our benchmark in assessing long reasoning capabilities. (2) GPT-4o (OpenAI, 2024) achieves the best performance on both multiple-choice and open-ended questions, with 56.01 accuracy (0-100) and 3.25 GPT4-score (0-5). There is also a large performance gap between GPT-4o and other open-source models which could be justified by the huge gap in the scale of the training data and GPUs used in training these models.(3) Qwen2-VL demonstrated the strongest performance among all open-source models, surpassing Gemini in MCQ skills by 1.4%. However, it struggled with open-ended tasks, where Gemini outperformed it by 1.06 GPT-score. One reason for Qwen2-VL's underperformance in open-ended tasks could be its reliance on training data that primarily consisted of videos without subtitles and images. As a result, it lacks the ability to fully comprehend video content that requires understanding both visual frames and audio (or subtitles). However Qwen2-VL is test with subtitles but it seams that it can't perform the alignment well.(4) short video models achieved the lowest performance because of information loss while sampling the long video into 8 or 45 frames in LLaVA-NeXT-Interleave (Li et al., 2024a) and MiniGPT4-video (Ataallah et al., 2024a)respectively. (5) As shown in Table 3, adding subtitles enables GPT-4 to achieve optimal performance. Subtitles allow the model to align characters, track their actions and clothing, and reason about specific events more effectively.Other models such as Qwen2-VL, InternVL2 and LLaVA-OneVision failed to exploit the subtitles. **Performance on specific skills.** Table 2 shows the performance of the SOTA long video understanding models on each skill. The performance varies significantly among different skills, highlighting the unique challenges introduced by each one. Observations of the results: (1) Character Actions is the most difficult MCQ question type, while Gemini achieving only 36.86% accuracy. The potential reason for the low performance is that

this question requires global reasoning across the entire hour-long video instead and the alignment between the audio or subtitles and the visual content. (2) All models struggle with Movie Spoiler questions in open-ended tasks, with the best model achieving a score of only 2.67 out of 5. The difficulty lies in the need for deeper understanding and reasoning to get the correct answer. Since Movie Spoiler questions are meaningful for human-centric video understanding, current model capabilities need improvement. (3) All models' achieved a good performance for the Local understanding questions. This shows that the main challenge for existing models is long-sequence global reasoning. (4) For the local questions our results is consistence with Gemini technical report (Gemini, 2024) that shows that Gemini and GPT-4o excel in the "needle in the haystack" skill, achieving high scores across all modalities. This aligns with our benchmark results in the local understanding skill, where Gemini and GPT-4o achieve the highest scores among all skills.

## 4.4 Effect of Subtitles

As shown in table 3 Incorporating subtitles intuitively enhances model performance; however, the extent of improvement varies across different models. For instance, GPT-4o demonstrates the most significant benefit from the inclusion of subtitles, while other models show minimal changes. This disparity suggests that GPT-4o effectively integrates and aligns multimodal inputs, leveraging both visual and textual modalities. In contrast, other models in the table trained exclusively on visual data lack the capacity to fully utilize the additional context provided by subtitles, thereby limiting their performance gains.

| Models | Frame Rate | Subtitles | Global Appearance | Scene Transitions | Character Actions | Chronological Understanding | Local Understanding | Summar-ization | Deep Context Understanding | Spoiler Understanding | Linking Events | Avg. Acc. | Avg. Score |
|---|---|---|---|---|---|---|---|---|---|---|---|---|---|
| Random Performance | – | – | 16.68 | 16.66 | 16.14 | 41.51 | 20.00 | N/A | N/A | N/A | N/A | 22.20 | N/A |
| GPT-4o | 250 | ✓ | 45.98 | 46.35 | 35.32 | 68.02 | 81.7 | 3.46 | 3.38 | 2.72 | 3.47 | 55.47 | 3.26 |
| GPT-4o | 250 | ✗ | 22.68 | 30.89 | 17.52 | 43.51 | 21.93 | 1.73 | 0.37 | 0.67 | 0.68 | 27.31 | 0.86 |
| Qwen2-VL | 250 | ✓ | 36.6 | 30.2 | 36.64 | 50.23 | 59.89 | 0.67 | 2.05 | 1.39 | 2.82 | 42.71 | 1.73 |
| Qwen2-VL | 250 | ✗ | 36.97 | 28.12 | 36.97 | 49.06 | 47.56 | 0.35 | 1.69 | 1.26 | 2.49 | 39.74 | 1.44 |
| LLaVA-OneVision | 128 | ✓ | 36.6 | 23.95 | 25.911 | 45.49 | 48.6 | 0.55 | 1.79 | 1.3 | 2.58 | 36.11 | 1.56 |
| LLaVA-OneVision | 128 | ✗ | 39.28 | 22.39 | 26.37 | 44.36 | 42.8 | 0.43 | 1.56 | 1.21 | 2.31 | 35.04 | 1.38 |
| InternVL2 | 128 | ✓ | 25.89 | 21.35 | 24.12 | 44.33 | 41.62 | 0.72 | 1.69 | 1.27 | 2.53 | 31.46 | 1.55 |
| InternVL2 | 128 | ✗ | 30.35 | 19.27 | 24.45 | 44.43 | 32.74 | 0.5 | 1.5 | 1.21 | 2.42 | 30.25 | 1.41 |

Table 3: Analysis of the impact of subtitles on model performance.

## 5 Conclusion

We introduced InfiniBench, a comprehensive benchmark for very long-form video understanding, featuring the longest average video duration (52.59 minutes) and the largest number of question-answer pairs (108.2K). Our diverse and human-centric questions evaluate nine distinct skills, posing significant challenges to current Multi-Modality Large language Models (MLMMs). Evaluations reveal that all existing models, including the commercial Gemini 1.5 Flash and GPT-4o and various open-source models, struggle with InfiniBench, particularly in tasks requiring deep context understanding and critical thinking. Despite these challenges, GPT-4o outperforms all open-source models across all skills. InfiniBench aims to bridge the gap in long-form video understanding benchmarks, promoting the development of LMMs toward achieving human-level comprehension and reasoning.

## 6 Limitations

This section outlines the limitations of our work: **Restricted Video Sources:** The video sources utilized in this study are limited exclusively to movies and television shows. Consequently, the benchmark lacks a broader spectrum of general videos encompassing various aspects of human life or the diverse field of wildlife. **Dependency on Transcripts:** The generation pipeline of questions and answers employed in this benchmark is inherently dependent on the availability of transcripts. This reliance confines its applicability to movies and television shows where such transcripts are readily available. For more general videos, the absence of transcripts poses a significant challenge, thereby limiting the pipeline's utility in those contexts. We hope to overcome these limitations in the future work.

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

# A  Human verification

To verify the reliability of our benchmark, we conducted a human evaluation to validate the correctness of our questions and answers pairs. Due to the challenging nature of our benchmark, which involves understanding videos over an hour long, we randomly sample 10% of the data for human verification.

More specifically, we asked nine annotators to read the questions related to each video and then watch the full video before validating the questions then start validating the questions. Each video averages 80 questions and is over an hour long, requiring 3 to 4 hours for annotators to verify. The study shows a great alignment between the human responses and our benchmark, where the accuracy of the true questions across different skills is **95.8%**. The detailed accuracy of the true questions per skill is reported in the table 4. The remaining two skills, i.e., Local understanding and sum-

| Skill Name | Number of Questions | Accuracy of Correctness (%) |
|---|---|---|
| Linking Events | 2297 | 98.00 |
| Character Actions | 667 | 94.90 |
| Deep Context Understanding | 2172 | 96.50 |
| Global Appearance | 135 | 89.62 |
| Scene Transitions | 103 | 88.34 |
| Spoiler Questions | 43 | 95.34 |
| Temporal Questions | 2927 | 94.08 |

Table 4: Human verification for all the GPT skills involved in the generation pipeline

marization questions, do not need human verification due to these annotaions are arleady humanly verified.For the quality of TVQA(Lei et al., 2019) questions and answers is well-documented in the TVQA paper (Lei et al., 2019) (Section 3.2, Table 8).It explicitly mentions that "The negative answers in TVQA are written by human annotators. They are instructed to write false but relevant answers to make the negatives challenging." This demonstrates that the questions and options are carefully crafted and verified by humans, ensuring their quality and relevance. for the scrapped summaries ,according to IMDB's contribution guidelines, the contributors must follow strict instructions when submitting plot summaries.and each contribution is reviewed and approved by the IMDB team before publication. This rigorous process ensures that IMDB summaries are reliable, accurate, and already human-verified.

# B  Ablations

## B.1  Effect of different input frames

To see the effect of different frame rates we conducted this ablation between three open-source models such as Qwen2VL, InternVL, LLaVA-OneVision and the best-performing model on our benchmark, GPT-4o, with different input frame rates. From Tab 5 we see that feeding more frames intuitively improves accuracy, but the degree of improvement varies across models. For instance, GPT-4o benefits the most from higher frame rates, while LLaVA-OneVision's performance remains almost unchanged despite using an 8x higher frame rate. The limited benefit of higher frame rates for LLaVA-OneVision may be attributed to its training strategy. LLaVA-OneVision is trained jointly on single images, multi-images, and videos. This strategy employs a balanced visual representation approach, aggressively down sampling video inputs to ensure parity with image-based scenarios. While effective for general tasks, this aggressive down sampling likely hurts LLaVA-OneVision's ability to understand long videos, limiting its benefit from higher frame rates. There are specific skills benefit more from higher frame rates. For example, the "local vision+text" skill improves most as it relies on short sequential shots. Increasing the frame rate reduces the chance of missing critical shots related to the answer, thereby boosting accuracy for such tasks. The results demonstrate that while higher frame rates generally improve performance, the degree of improvement depends on the model's design and training strategy. Models like GPT-4o, optimized for sequential inputs,

show significant gains, whereas models like LLaVA-OV, which aggressively downsample videos, see minimal benefits.

| Models | #Frames | Subtitles | Global Appearance | Scene Transitions | Character Actions | Chronological Understanding | Local Understanding | Summar-ization | Deep Context Understanding | Spoiler Understanding | Linking Events | Avg. Acc. | Avg. Score |
|---|---|---|---|---|---|---|---|---|---|---|---|---|---|
| Random Performance | – | – | 16.68 | 16.66 | 16.14 | 41.51 | 20.00 | N/A | N/A | N/A | N/A | 22.20 | N/A |
| GPT-4o | 250 | ✓ | 45.98 | 46.35 | 35.32 | 68.02 | 81.70 | 3.46 | 3.38 | 2.72 | 3.47 | 55.47 | 3.26 |
| GPT-4o | 128 | ✓ | 18.98 | 29.84 | 17.92 | 43.12 | 22.10 | 1.78 | 0.37 | 0.61 | 0.69 | 26.39 | 0.86 |
| GPT-4o | 16 | ✓ | 20.37 | 31.93 | 16.38 | 42.32 | 20.22 | 1.68 | 0.35 | 0.63 | 0.65 | 26.24 | 0.83 |
| Qwen2-VL | 250 | ✓ | 36.60 | 30.20 | 36.64 | 50.23 | 59.89 | 0.67 | 2.05 | 1.39 | 2.82 | 42.71 | 1.73 |
| Qwen2-VL | 128 | ✓ | 32.58 | 28.64 | 34.59 | 49.33 | 54.98 | 0.59 | 1.87 | 1.31 | 2.75 | 40.02 | 1.63 |
| Qwen2-VL | 16 | ✓ | 30.80 | 20.83 | 32.20 | 46.59 | 42.90 | 0.30 | 1.53 | 1.16 | 2.44 | 34.66 | 1.36 |
| LLaVA-OneVision | 128 | ✓ | 36.60 | 23.95 | 25.91 | 45.49 | 48.60 | 0.55 | 1.79 | 1.30 | 2.58 | 36.11 | 1.56 |
| LLaVA-OneVision | 16 | ✓ | 41.51 | 24.47 | 25.97 | 44.27 | 40.15 | 0.48 | 1.48 | 1.33 | 2.30 | 35.27 | 1.40 |
| InternVL | 128 | ✓ | 25.89 | 21.35 | 24.12 | 44.33 | 41.62 | 0.72 | 1.69 | 1.27 | 2.53 | 31.46 | 1.55 |
| InternVL | 16 | ✓ | 23.21 | 20.83 | 25.18 | 44.82 | 31.95 | 0.70 | 1.54 | 1.28 | 2.60 | 29.20 | 1.53 |

Table 5: Impact of varying frame rates on model performance.

## B.2 IMPACT OF THE "I DON'T KNOW" OPTION

The results in Table 6 indicate that excluding the "I don't know" option improves the model's performance metrics. This increase occurs because, in the absence of this option, the model is compelled to select a response, even when it lacks sufficient evidence, thereby increasing the likelihood of correctly guessing through chance. Conversely, including the "I don't know" option introduces an additional challenge, as it requires the model to explicitly acknowledge uncertainty when it cannot confidently match a provided answer.

This option also serves as a valuable diagnostic tool for assessing hallucination tendencies. Specifically, when evaluating with GPT-4o, the "I don't know" option allows for the detection of instances where the model generates an answer inconsistent with the given evidence. GPT-4o is designed to utilize this option when the predicted response either lacks sufficient evidence to align with one of the provided choices or constitutes a hallucination. Thus, the inclusion of the "I don't know" option enhances the robustness of model evaluation by accounting for uncertainty and mitigating the impact of overconfident, unsupported predictions.

| Models | # Frames | Subtitles | I don't know option | Global Appearance | Scene Transitions | Character Actions | Chronological Understanding | Avg. Acc. |
|---|---|---|---|---|---|---|---|---|
| Random Performance | – | – | – | 16.68 | 16.66 | 16.14 | 41.51 | 22.75 |
| Qwen2-VL | 250 | ✓ | ✓ | 36.60 | 30.20 | 36.64 | 50.23 | 38.42 |
| Qwen2-VL | 250 | ✓ | ✗ | 36.16 | 33.85 | 39.36 | 48.59 | 39.49 |
| InternVL2 | 128 | ✓ | ✓ | 25.89 | 21.35 | 24.12 | 44.33 | 28.92 |
| InternVL2 | 128 | ✓ | ✗ | 29.46 | 22.91 | 26.70 | 44.57 | 30.91 |
| LLaVA-OneVision | 128 | ✓ | ✓ | 36.60 | 23.95 | 25.91 | 45.49 | 32.99 |
| LLaVA-OneVision | 128 | ✓ | ✗ | 38.83 | 23.43 | 27.50 | 46.05 | 33.95 |

Table 6: Analysis of the impact of incorporating the "I don't know" option on model performance.

## C DEEP DATA EXPLORATION

### C.1 ANALYSIS OF DATA LEAKAGE

To genuinely assess the data leakage, we deliberately drop the video and only feed the question and some context about the episode or the movie without any visual inputs.

For instance, here is the input prompt in the blindness case:

"This is a question for a video from show season_num episode_num, use your knowledge to answer this question: question"

We have conducted the blindness experiments using two models, Qwen and GPT-4o. As shown in the tables 7, in most skills, the blind models' performance is too close to the random performance.

For instance, on the "global appearance" and the "scene transitions" skills, Qwen achieves 19.6 and 21, while GPT-4o achieves 20.8 and 22.5, approximately equal to the random performance of around 17 for both skills.

In contrast, only the blind Qwen on one skill, the "Character Actions", achieves closer performance than the Qwen, which takes the visual input, 36.6 and 36, respectively. This could be interpreted

as the model using its common sense to answer the question. The choices in this skill contain valid actions, and only their order is wrong. Thus, we argue that the model could perform well using common sense to order the events. To test our hypothesis, we assess the model performance on this skill as an open-ended question without choices. We leverage GPT-4o to score the models' outputs out of 5, where 0 is the worst and 5 is the best. The detailed prompt used while scoring is depicted in Figure 7. As expected, when we remove the visual input, the accuracy drops significantly from 0.79 to 0.003 as shown in the table below.

| Models | Global Appearance | Scene Transitions | Character Actions | Chronological Understanding | Local Understanding | Summar- ization | Deep Context Understanding | Spoiler Understanding | Linking Events | Avg. Acc. | Avg. Score |
|---|---|---|---|---|---|---|---|---|---|---|---|
| Random Performance | 16.68 | 16.66 | 16.14 | 41.51 | 20.00 | N/A | N/A | N/A | N/A | 22.20 | N/A |
| GPT-4o Video + sub + question | 45.98 | 46.35 | 35.32 | 68.02 | 81.70 | 3.46 | 3.38 | 2.72 | 3.47 | 55.47 | 3.26 |
| GPT-4o Question + Video Info | 20.83 | 22.51 | 17.18 | 42.82 | 17.17 | 1.70 | 0.37 | 0.68 | 0.70 | 24.10 | 0.86 |
| GPT-4o Question | 14.81 | 24.08 | 15.78 | 42.35 | 16.44 | 1.75 | 0.36 | 0.67 | 0.67 | 22.69 | 0.86 |
| Qwen Video + sub + question | 36.60 | 30.20 | 36.64 | 50.23 | 59.89 | 0.67 | 2.05 | 1.39 | 2.82 | 42.71 | 1.73 |
| Qwen Question + Video Info | 19.64 | 21.35 | 35.05 | 46.57 | 39.71 | 0.28 | 1.70 | 1.48 | 2.60 | 32.46 | 1.51 |
| Qwen Question | 18.75 | 19.27 | 29.62 | 45.49 | 38.29 | 0.00 | 0.97 | 0.76 | 1.7 | 30.28 | 0.86 |

Table 7: Analysis of the impact of different input modalities on the performance of Qwen2VL and GPT-4o.

| Input | GPT-4o score |
|---|---|
| Qwen2VL Video+Questions | 0.79 |
| Qwen2VL only Question | 0.003 |

Table 8: Evaluation of Qwen2-VL on character action recognition as an open-ended skill without multiple-choice options.

### C.2 CHECK QUESTIONS DUPLICATION

To ensure the dataset is free from (near) duplicate questions, we implemented the following approach:

Encoding and Similarity Calculation: We used M3-Embedding (Chen et al., 2024) to encode the questions and answer choices into vector representations.the cosine similarity was then calculated to identify potential duplicates. To account for varying degrees of similarity, we evaluated three thresholds: 90%, 95%, and 98% cosine similarity. As shown in Figure 6 the vast majority of questions across all skills are unique, with no duplicates detected. Only two skills, Temporal Questions and Character Actions, showed instances of potential duplicates. Upon investigation, we found that the duplicates detected were false positives. For example, the following pair of questions was flagged as duplicates because they differ only by the words "before" and "after". However, this small difference completely changes the meaning of the question. for instance : Q1: Did the event flashback to Phoebe completing a mile on a hippity-hop before turning thirty, happen before the event Monica makes breakfast with chocolate-chip pancakes?

Q2: Did the event flashback to Phoebe completing a mile on a hippity-hop before turning thirty, happen after the event Monica makes breakfast with chocolate-chip pancakes? These examples highlight the importance of semantic context in evaluating the similarity of questions, as minor lexical differences can significantly alter meaning. Based on our analysis, we are confident that the dataset is free from true duplicates, and our pipeline effectively identifies and handles potential near-duplicates. The false positives flagged by the similarity detection process underscore the complexity of semantic evaluation, especially in nuanced question construction.

### C.3 TRANSCRIPT VS. SUBTITLES

As shown in Figure ?? that shows the difference between subtitles and the transcript. Transcripts are detailed documents created by movie or TV show writers. They provide comprehensive information beyond spoken dialogue, including: scene descriptions, context about settings, locations,character actions ,and camera angles or shot compositions. Transcripts serve as blueprints for visual and narrative elements, helping to extract visual insights and design challenging, reliable benchmark questions.The Key Point: Transcripts are used only during the benchmark creation process to ensure robustness and question diversity, not during inference or evaluation.

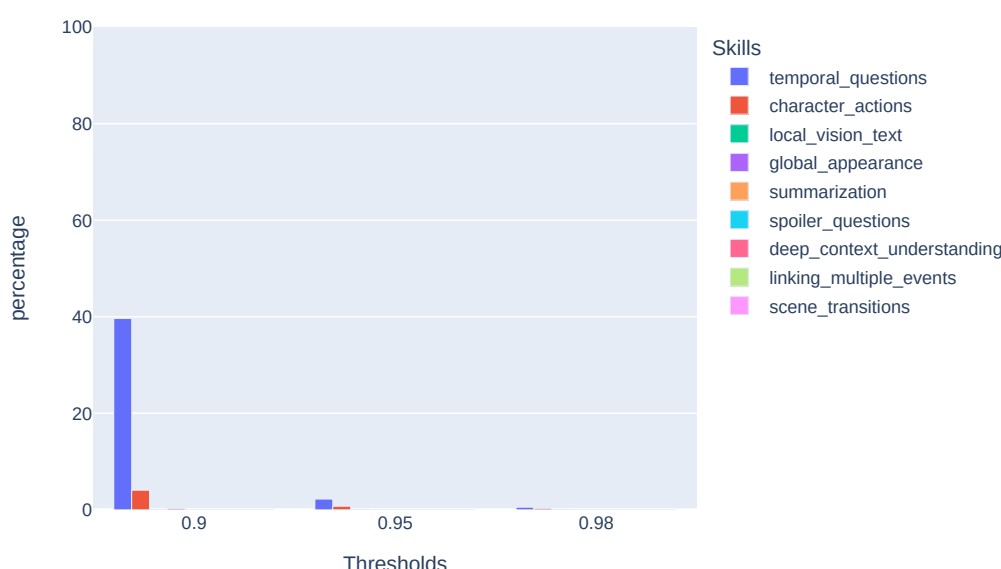

Figure 6: InfiniBench duplication for different thresholds using cosine similarity of text vector embeddings

| Skill Name | Number of options | | | | Weighted |
|---|---|---|---|---|---|
| | 2 | 5 | 6 | 7 | Random accuracy |
| Global Appearance | 0 | 9 | 1447 | 0 | 16.68 |
| Scene transitions | 0 | 0 | 920 | 0 | 16.66 |
| Character actions | 0 | 0 | 5829 | 1665 | 16.14 |
| temporal order of events | 24056 | 0 | 8208 | 0 | 41.51 |
| Local vision + text questions | 0 | 15246 | 0 | 0 | 20.00 |

Table 9: Detailed calculations for the random accuracy for the whole MCQ skills

Subtitles focus solely on translating spoken dialogue into text, typically extracted by transcribing the video's audio. The subtitles are optional input for the AI model during inference, representing an additional modality when available. During inference, the model's input consists of video frames and, optionally, subtitles.

## C.4 WEIGHTED RANDOM ACCURACY

Table. 9 provides details about the number of options for each multiple-choice question (MCQ) skill, including Global Appearance, Scene Transitions, Sequence of Character Actions, Temporal Order of Events, and Local Vision and Context Questions. The table also reports the weighted random accuracy for each skill.

**Transcript of episode 1 season 1 of Friends TV shows:**

[Scene: Central Perk, Chandler, Joey, Phoebe, and Monica are there.]
Monica: There's nothing to tell! He's just some guy I work with!
...
(Ross gestures his consent.)
Joey: Strip joint! C'mon, you're single! Have some hormones!
(Rachel enters in a wet wedding dress and starts to search the room.)

**Subtitle of the same episode**
1
00:00:54,012 --> 00:00:57,641
There's nothing to tell.
It's just some guy I work with.

2
00:00:57,892 --> 00:00:59,962
There's gotta be something wrong with him.

Figure 7: Different between transcript and subtitles

## D    EVALUATION DETAILS

### D.1    EVALUATION METRIC DETAILS

For MCQs, large language models (LLMs) do not consistently provide direct responses. The output may vary, sometimes giving the option number, other times the option sentence, or occasionally providing additional clarifications for the selected option. For example, an LLM might produce a response such as: "I think option 1 is close, but my final answer will be option 2." Additionally, some responses may include hallucinations not found in the given options. To address this variability, we implemented a standardized evaluation method using GPT-4o to match the LLM's prediction with one of the provided options. Specifically, we input the set of options and the LLM's prediction into GPT-4o, which then attempts to match the predicted answer with one of the given options. If no matching option is found or if the response includes hallucinations, GPT-4o matches the prediction with an "I don't know" option. Using the prediction option number and the ground truth option number, we then calculate the accuracy. For open-ended questions, GPT-4o assessed the LLMs' predictions based on several criteria: correctness, meaningfulness, alignment with the expected answer, presence of hallucinations, and completeness. Using these criteria, GPT-4o assigned a score from 0 to 5 to indicate the overall quality of each response.

### D.2    EVALUATION PROMPTS DETAILS

In this section we will discuss the details for the prompts that have been used for evaluation for both the open ended questions and multiple choices. Figure. 8 show the detailed prompt used for the results matching.Figure 9 show the detailed prompt for the GPT-4o scores.

# E    EXTRA BENCHMARK EXAMPLES

Here in this sections, we are showing more examples of our benchmark skills such as the temporal order of events in Fig. 13, linking multiple events in Figure.14, deep context understanding in Figure. 12 , local questions in Figure.15 ,spoiler questions in Figure.17, Sequence of character actions Figure.18, and summarization in Figure. 16.

# F    SUCCESS AND FAILURE CASES

In this section, we present examples of both success and failure cases in question generation using GPT-4o. Figure 20 illustrates cases involving the generation of Temporal Order of Events questions, while Figure 19 showcases examples related to Linking Multiple Events questions. As highlighted in the human evaluation section A, such failure cases are infrequent, with 92% of the generated data verified as accurate.

# G    QUALITATIVE RESULTS

In this section, we present qualitative results to assess how the evaluated models perform in answering the benchmark questions. We also examine how GPT-4o scores these responses compared to the ground truth, particularly in the case of open-ended questions. Figure 21 shows an example of the deep context understanding skill , Figure 22 shows an example of Global appearance skill, Figure 23 shows an example of the Scene transition skill and Figure 24 shows an example of the spoiler questions skill.

in the spoiler questions and deep context understanding , we can see the GPT-4o scores for each answer.

# H    INFINIBENCH GENERATION DETAILS

This section elaborates on the specific prompts employed to generate questions for each skill category. The prompts, utilized within the GPT-4o framework, are depicted in Figures 25, 27, 26, 28,29.These figures provide the exact phrasing and structure used for question generation, ensuring reproducibility and clarity in the benchmarking creation process.

## MCQ matching prompt:

**System prompt:**
You are an intelligent chatbot designed to evaluate the correctness of generative outputs for multiple-choice questions (MCQs).
Your task is to match the predicted answer with one of the provided options, which include an 'I don't know' option. If there is no match between the predicted answer and the options, choose the option that says, 'I don't know'. Here's how you can accomplish the task:
------
## INSTRUCTIONS:
- Focus on finding a meaningful match between the predicted answer and the correct option.
- Consider synonyms or paraphrases as valid matches.
- Choose an option only if you believe there is sufficient evidence to directly derive the answer from the predicted information or indirectly with minimal reasoning. If there isn't enough evidence to support any option, simply select the option with 'I don't know.'
- Provide only the integer that represents the option number for your evaluation decision.
- Evaluate as a human would, considering context and meaning, not just exact words.
- Provide your answer in the form of a Python dictionary string with the key 'decision', such as {'decision': 3}.
**User prompt:**
Please evaluate the following question-answer pair:
Options: {options}
Predicted Answer: {pred}
Provide your evaluation as a decision with the matched option number.
Generate the response in the form of a Python dictionary string with the key 'decision'.
DO NOT PROVIDE ANY OTHER OUTPUT TEXT OR EXPLANATION. Only provide the Python dictionary string.
For example, your response should look like this: {'decision': 1}.
Do not include any other information in your response such as ```python```.

Figure 8: Detailed prompt for MCQ evalaution

## Scoring evaluation prompt:

**System prompt:**
You are an intelligent chatbot designed to evaluate the correctness of generative outputs for question-answer pairs.
Your task is to compare the predicted answer with the correct answer and determine if they match meaningfully. Here's how you can accomplish the task:
------
## INSTRUCTIONS:
- Focus on the meaningful match between the predicted answer and the correct answer.
- Consider synonyms or paraphrases as valid matches.
- Evaluate the correctness of the prediction compared to the answer.
- Provide a score between 0 and 5, where 5 indicates the highest meaningful match.
- Penalize the score if the predicted answer contains hallucinations or is missing key parts of the correct answer.
- Assign your score based on how far the predicted answer is from the correct answer.
- Evaluate as a human would, not as a machine.
- Provide your score in the form of a Python dictionary string with the key 'score', such as {'score': 3.7}.
**User prompt:**
Please evaluate the following video-based question-answer pair:
Question: {question}
Correct Answer: {answer}
Predicted Answer: {pred}
Provide your evaluation only as a score where the score is an integer value between 0 and 5, with 5 indicating the highest meaningful match.
Generate the response in the form of a Python dictionary string with the key 'score'.
DO NOT PROVIDE ANY OTHER OUTPUT TEXT OR EXPLANATION. Only provide the Python dictionary string.
For example, your response should look like this: {'score': 4}.
Do not include any other information in your response such as ```python ```.

Figure 9: Detailed prompt for Scoring system evaluation

## Temporal questions:

**Q: Choose the correct option for the following question: Looking at these events : [Chandler reluctantly agrees to return to his old job after negotiation, Monica gets disappointed by the lost job opportunity, Phoebe gives Steve a painful massage as payback, Monica's audition dinner is ruined by Steve being stoned], how do they unfold in the episode?**

Option 1: [Phoebe gives Steve a painful massage as payback ,Chandler reluctantly agrees to return to his old job after negotiation, Monica gets disappointed by the lost job opportunity ,Monica's audition dinner is ruined by Steve being stoned],
Option 2: [Monica's audition dinner is ruined by Steve being stoned ,Chandler reluctantly agrees to return to his old job after negotiation, Phoebe gives Steve a painful massage as payback ,Monica gets disappointed by the lost job opportunity],
Option 3: I don't know,
Option 4: [Monica gets disappointed by the lost job opportunity ,Chandler reluctantly agrees to return to his old job after negotiation, Phoebe gives Steve a painful massage as payback ,Monica's audition dinner is ruined by Steve being stoned],
Option 5: [Chandler reluctantly agrees to return to his old job after negotiation ,Monica's audition dinner is ruined by Steve being stoned, Monica gets disappointed by the lost job opportunity ,Phoebe gives Steve a painful massage as payback]

00:00                                                                  22:46
                                                                       Time

Figure 10: Example for the temporal order of events skill

# Global Appearance

**Q: Choose the correct option for the following question:**
**Can you track the sequence of Penny's outfit changes in this episode?**

Option 1: gray tank top over a pink sports bra , mustard yellow vest over a white blouse ,
        red and pink floral dress with a blue tank top underneath , red floral dress over a
        red shirt , blue tank top

Option 2: red floral dress over a red shirt ,gray tank top over a pink sports bra ,
        red and pink floral dress with a blue tank top underneath mustard yellow vest over
        a white blouse , blue tank top

Option 3: black yoga pants and purple sports bra , white sundress ,black formal suit , black running
        shorts and a white tank top.
Option 4: I don't know.
Option 5: blue tank top , red and pink floral dress with a blue tank top underneath ,
        red floral dress over a red shirt , gray tank top over a pink sports bra , mustard yellow vest
        over a white blouse

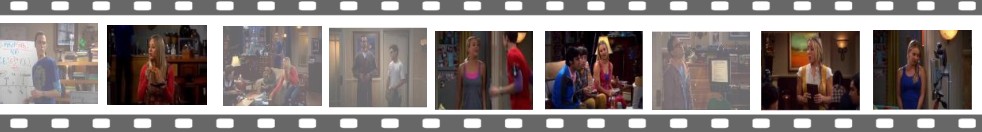

Figure 11: Example for the Global appearance skill.

## Deep context understanding:

**Q: What does Celia do when Marcel Ross's monkey starts interacting with her during the date?**

Celia screams and is unable to handle Marcel pulling at her hair until Ross lifts Marcel away.

00:00                                                                                      22:46
                                                                                           Time

Figure 12: Example for the deep context understanding skill

# Scenes Transition

**Q :Choose the correct option for the following question:**
**What is the chronological order of scenes in this episode?**

Option 1 : Monica and Rachel's apartment , Madison Square Garden, Duncan's dressing room ,The street , Ross's apartment, Central Perk , Madison Square Garden , Ross's apartment, Outside in the hallway

Option 2 : Ross's apartment , The street , Madison Square Garden, Duncan's dressing room , Monica and Rachel's apartment , Madison Square Garden , Central Perk , Ross's apartment, Outside in the hallway

Option 3 : I don't know,

Option 4 : Ross's apartment, Outside in the hallway , Madison Square Garden, Duncan's dressing room , The street , Central Perk , Madison Square Garden , Ross's apartment , Monica and Rachel's apartment

Option 5 : Monica and Rachel's apartment , Central Perk , Madison Square Garden , Ross's apartment , Madison Square Garden, Duncan's dressing room , Ross's apartment, Outside in the hallway , The street

Figure 13: Example for the Scenes transitions skills.

## Linking multiple events :

**Q: What is the connection between Monica's failed dinner and Phoebe's reaction during Steve's next massage appointment?**

Monica's dinner for Steve fails due to his stoned condition and disruptive behavior. Phoebe, out of frustration with Steve's behavior and the ruined dinner, takes out her anger on him during his next massage appointment by giving him a painful massage.

00:00                                                                      22.47
                                                                            Time

Figure 14: Example for the linking multiple events skill

## Local questions

> **Q: Choose the correct option for the following question:**
> **What is Joey eating when Chandler is on the phone with the guy from his old job?**

Option 1: A piece of pie
Option 2: Popcorn
Option 3: A donut
Option 4: A bread roll
Option 5: A slice of pizza

00:00                                                                    22.47
                                                                         Time

Figure 15: Example for the local questions

## Summarization

> **Q: Please summarize the video with as much detail as possible.**

Monica cooks a gourmet meal for Steve (Jon Lovitz), a restaurateur looking for a new head chef. Steve is a massage client for Phoebe, and she makes the introduction between Monica and him. The job is perfect as Steve wants something eclectic and needs someone who can create the entire menu. As an audition, Monica is cooking dinner for him the coming week. She wants Phoebe to be there. Monica hires a professional waitress Wendy (for $10/hr.), which offends Rachel (Monica says that she needed a professional waitress).Wendy bails on Monica at the last minute. Monica begs Rachel and even says that she gave her shelter when she had nowhere else to go.. Eventually she offers Rachel $20/hr. He arrives stoned and wants to eat everything in sight, including taco shells and gummy bears. Phoebe tells Rachel who tries to handle the situation by offering Steve some wine. Eventually Monica realizes that Steve is super stoned. She tries to yank the gummy bears from Steve, and they end up falling in the punch bowl.. Dinner is a total disaster, and the gang tells her that she doesn't want to work for a guy like that. After working as a data processor for five years, Chandler gets promoted to supervisor. Chandler quits, claiming he only intended for his job to be temporary (and Chandler already has been there for over 5 yrs.). Chandler goes to meet a career counselor. After 8 hrs. of aptitude, personality and intelligence tests he learns that he is fit for a career in data processing, for a large multinational corporation. he is disappointed as he always pictured himself doing something cool. When his boss calls and offers more money (& more bonus.. Chandler resists, but the boss keeps throwing more and more numbers), Chandler caves and goes back to work. Chandler gets the corner office, and he shows it off to Phoebe. He has a view and an assistant. But Chandler has more responsibility now and starts spending more time & late nights at work and yelling at his juniors. He doesn't like it. Ross has a date with a beautiful colleague named Celia (Melora Hardin) (curator of insects at the museum) and gives new meaning to the term 'spanking the monkey' when she meets Marcel. The date goes bad when Marcel hands on Celia's hair and pulls it. Eventually Ross takes Celia to bed, and she wants him to talk dirty and he says 'Vulva'. Ross turns to Joey for advice as Celia wants him to talk dirty as foreplay. Joey gets Ross to practice on him.. When Ross talks smack, Chandler overhears and amuses himself at their expense. Ross does well at the next date and talks very dirty (with theme, plot, motif and story-lines. at one point there were villagers), but eventually they get tired and cuddle. Phoebe takes out her anger at Steve at his next massage appointment by treating him to a bad massage (she elbows him on his back and pinches his skin so that it hurts).

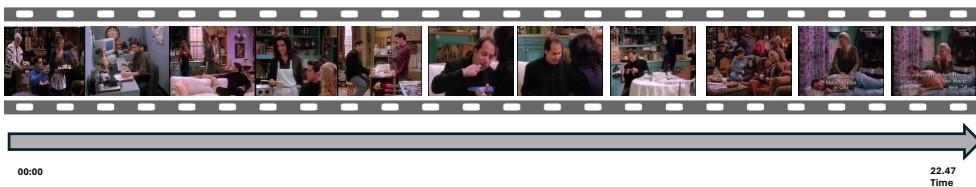

00:00                                                                    22.47
                                                                         Time

Figure 16: Example for the summarization skill

## Spoiler questions

**Q: Why didn't the Arquillian in the jeweler's head simply tell Jay that the galaxy was on his cat's collar?**

To add a bit of mystery to the story. If he'd said 'the galaxy in the jewel on the cat's collar', the movie would have ended much faster. Actually, Arquillian was indeed trying to tell Jay that the galaxy was on the cats collar. He just didn't have the correct vocabulary to do so. Note how he stumbles over the word \"war\". He almost certainly thinks \"belt\" is the correct word for \"collar\", which is understandable because the articles of clothing are identical, as the only differences are that one is worn around the waist and the other is worn around the neck. And the cat's name is Orion, so he's being accurately descriptive, not deceitful.It's likely that the Arquillian didn't understand much English and that the Jeweler's body had a translator in it when conversing with humans. It was likely damaged when Edgar stabbed it through the neck.

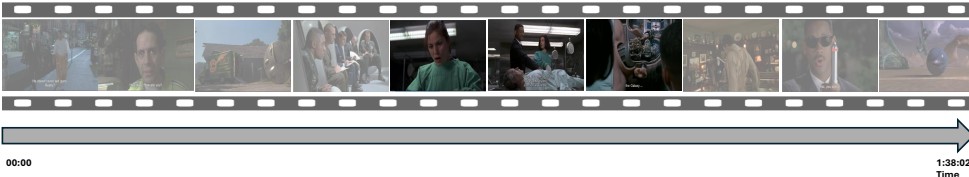

00:00                                                                 1:38:02
                                                                      Time

Figure 17: Example for the spoiler questions

## Character Actions :

**Q: Choose the correct option for the following question: What did Rachel do through this video?**

**Option 1:** enjoying a cappuccino with a dash of cinnamon at a trendy coffee shop , discussing the latest book club read with a friend over lunch , savoring a slice of gourmet pizza with sun-dried tomatoes and arugula at a pizzeria , exploring an art museum's new exhibit on modern sculpture.

**Option 2:** I don't know

**Option 3:** Rachel attends an initial interview, accidentally kisses Mr. Zelner. , She calls back, gets another meeting with Mr. Zelner and eventually gets the job after apologizing and explaining her actions. , She practices handshaking with Phoebe and Monica., Accidentally touches Mr. Zelner's crotch while offering a handshake., Rachel gets a second interview call and gets ink on her lips during the second attempt., Rachel enters Central Perk and announces her job interview at Ralph Lauren., Rachel goes home, realizes her mistake regarding the ink., Misinterprets Mr. Zelner's gesture and walks out thinking he's making an advance.

**Option 4:** Rachel enters Central Perk and announces her job interview at Ralph Lauren., She practices handshaking with Phoebe and Monica. , Rachel attends an initial interview, accidentally kisses Mr. Zelner. , Rachel gets a second interview call and gets ink on her lips during the second attempt. , Misinterprets Mr. Zelner's gesture and walks out thinking he's making an advance. , Rachel goes home, realizes her mistake regarding the ink. , She calls back, gets another meeting with Mr. Zelner and eventually gets the job after apologizing and explaining her actions. , Accidentally touches Mr. Zelner's crotch while offering a handshake.

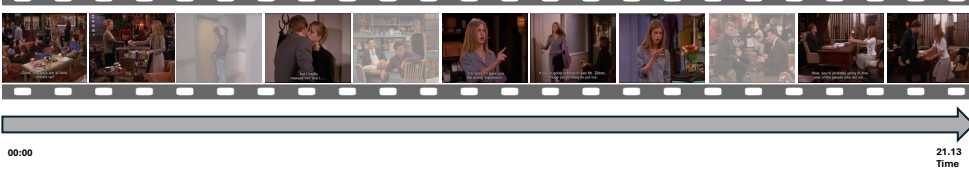

00:00                                                                 21.13
                                                                      Time

Figure 18: Example for sequence of character actions questions

## Linking multiple events

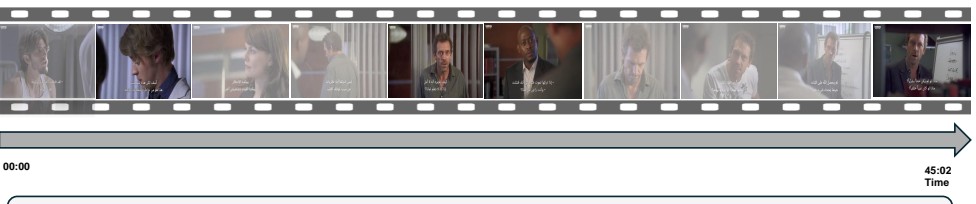

00:00                                                           45:02
 Time

How does Dr. House's internal conflict towards the end connect to the events of the episode?

Dr. House's internal conflict at the end ties together the various events of the episode. The stress of the almost Sci-fi case, the emotional impact of Foreman's departure, and the unresolved medical mystery all contribute to House's turmoil, leaving him with an immense conflict that sets the stage for the next season.

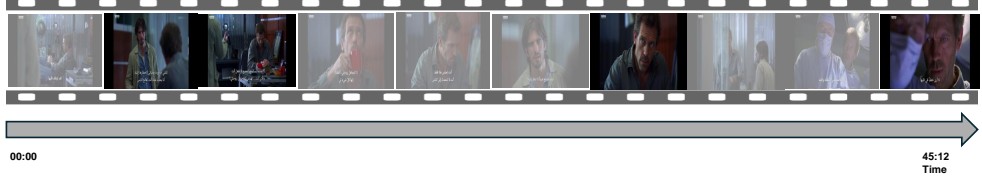

00:00                                                           45:12
 Time

In what ways do the sea rescue and the medical mystery serve as catalysts for character development within the Diagnostics team?

The sea rescue brings the couple to the team's attention, setting off a series of events that act as catalysts for character development. The challenging medical mystery forces team members to confront their own abilities, resolve conflicts, and cope with Foreman's departure, leading to significant personal and professional growth.

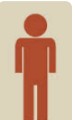

Why the answer is not valid ?

Because the couple traveled a great distance, time and danger to reach the hospital through the Coast Guards. This encourages the House team to do their best for this case.

Figure 19: Examples of success and failure cases in Linking Multiple Events questions.

**Temporal questions**

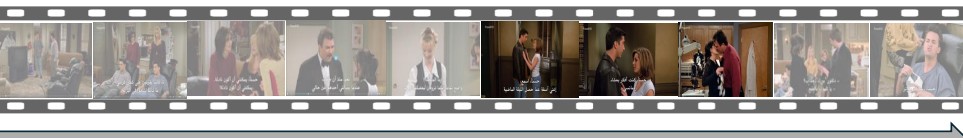

Choose the correct option for the following question:  Given the events listed ['Rachel apologizes to Ross and suggests a romantic dinner to make it up to him', 'Monica goes to her eye appointment with Dr. Burke', 'Monica and Dr. Burke kiss'], what is the sequential order in this episode?

['Rachel apologizes to Ross and suggests a romantic dinner to make it up to him', 'Monica goes to her eye appointment with Dr. Burke', 'Monica and Dr. Burke kiss']

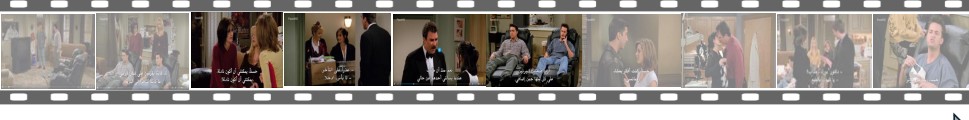

Choose the correct option for the following question: Looking at these events ['Phoebe offers to waitress for Monica instead of Rachel', 'Monica and Phoebe arrive at Dr. Burke's apartment for a catering job', 'Dr. Burke tells Monica about his divorce', 'Chandler orders a pizza for him and Joey'], how do they unfold in the episode?

['Phoebe offers to waitress for Monica instead of Rachel', 'Monica and Phoebe arrive at Dr. Burke's apartment for a catering job', 'Chandler orders a pizza for him and Joey', 'Dr. Burke tells Monica about his divorce']

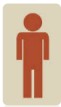

Why the answer is not valid ?

Because Dr. Burke tells Monica about his divorce' before 'Chandler orders a pizza for him'

Figure 20: Examples of success and failure cases in Temporal Order of Events questions.

## Deep context understanding:

**Q: What does Celia do when Marcel Ross's monkey starts interacting with her during the date?**

GT: Celia screams and is unable to handle Marcel pulling at her hair until Ross lifts Marcel away.

GPT-4o: Celia becomes uncomfortable and asks for help when Marcel, Ross's monkey, starts interacting with her by grabbing her hair.

Gemini: When Marcel starts interacting with Celia during the date, she gets nervous and tells Ross that she has to leave.

Llama-vid: Celia is frightened by Marcel Ross's monkey's aggressive behavior and runs out of the apartment in terror.

Moviechat: In the video, we see a woman sitting at a table in a home. The man wearing a black shirt starts talking to her, and she reacts by talking to him back. Then, he puts his hand on the table and leans in towards her. She does not respond and just looks back at him. Finally, the man in the black shirt walks away from the table to talk to someone else, and the woman remains sitting there.

Large world model: When Marcel's monkey starts interacting with Celia during the date, Celia becomes very serious and focused on her own monkey. She does not appreciate any disturbances and prefers to have uninterrupted interactions with her pet.

Figure 21: Qualitative results of deep context understanding questions

## Global appearance:

**Q: Choose the correct option for the following question: Can you track the sequence of Penny's outfit changes in this episode?**

GT: Option 1: red floral dress over a red shirt ,gray tank top over a pink sports bra , red and pink floral dress with a blue tank top underneath mustard yellow vest over a white blouse , blue tank top.

GPT-4o : option 0

Gemini :**Option 1:** ['red floral dress over a red shirt', 'gray tank top over a pink sports bra', 'red and pink floral dress with a blue tank top underneath', 'mustard yellow vest over a white blouse', 'blue tank top']"

Llama-vid : "0"

Moviechat: Next, a man and woman are seen talking while standing in the bathroom. The man is wearing glasses, and they are talking at the black mirror in the bathroom. A group of people are sitting in a red chair in front of the fireplace. The man is wearing glasses and is standing behind the white couch. A group of people are seen inside the house, including a woman wearing a purple dress. A man is seen in front of the television in the living room. Finally, the man and woman are sitting in a chair in front of the TV. The TV has a white case and a black frame. The man is wearing a pink shirt and the woman is wearing a blue dress.

Large world model : Yes, in the video, the man who is talking to Penny is seen putting on a black vest. Later, he is seen putting on a red and white shirt, and then he is seen putting on a red and white vest.

Figure 22: Qualitative results of global appearance skill

**Scene transitions:**

Q :Choose the correct option for the following question: What is the chronological order of scenes in this episode?

Option 4 : Monica and Rachel's apartment , Central Perk , Madison Square Garden , Ross's apartment , Madison Square Garden, Duncan's dressing room , Ross's apartment, Outside in the hallway , The street

GPT-4o: option 0

Gemini : **Answer:** option 5: ["Ross's apartment", "Ross's apartment, Outside in the hallway", "Madison Square Garden, Duncan's dressing room", "Monica and Rachel's apartment", 'Madison Square Garden', 'The street', 'Central Perk']

Llama-vid : 1. Ross and Julie are in Ross' apartment.\n2. Ross and Julie are on their way to Madison Square Garden.\n3. Ross and Julie are in Duncan's dressing room at Madison Square Garden.\n4. Rachel and Monica are in their apartment.\n5. Rachel and Monica are at Central Perk.\n6. Rachel and Monica are in their apartment again.

Moviechat: First, we see a group of friends gathering around a bar in a New York apartment. A young man is seen with his hand on a woman's leg. Next, the group is seen sitting around a table in the apartment. Then, a man and a woman are seen in a dark room with a green table and a dark room with a man in a white shirt. The group is gathered around the table in the first scene, and then the men and the woman are seen in the dark room. They are then joined by another man in the green room. Lastly, the group is seen in a dark room with a green table. There is a man and a woman sitting on a couch, and the woman is crying. The group is again seen in a green room with a couch, and a woman is sitting on it. They are also seen in a dark room with a green table and a man and a woman sitting in a green room. The group is seen in a green room with a couch, and a woman is crying. Finally, the group is seen in a dark room with a green table and a man and a woman sitting in a green room.

Large world model :Yes, the scene changes from Monica and Rachel sitting on the floor to a man standing in front of them. The man then proceeds to cut his hair with a knife, and Monica and Rachel start to panic. The man then uses a comb to brush his hair, and the three of them start to panic again.

Figure 23: Qualitative results of scenes transition skill

**Spoiler questions**

00:00           1:38:02 Time

Q: Why didn't the Arquillian in the jeweler's head simply tell Jay that the galaxy was on his cat's collar?

To add a bit of mystery to the story. If he'd said, 'the galaxy in the jewel on the cat's collar', the movie would have ended much faster. Actually, Arquillian was indeed trying to tell Jay that the galaxy was on the cat's collar. He just didn't have the correct vocabulary to do so. Note how he stumbles over the word "war". He almost certainly thinks "belt" is the correct word for "collar", which is understandable because the articles of clothing are identical, as the only differences are that one is worn around the waist and the other is worn around the neck. And the cat's name is Orion, so he's being accurately descriptive, not deceitful. It's likely that the Arquillian didn't understand much English and that the Jeweler's body had a translator in it when conversing with humans. It was likely damaged when Edgar stabbed it through the neck.

GPT-4o: The Arquillian in the jeweler's head didn't simply tell Jay that the galaxy was on his cat's collar because he was using cryptic and indirect communication to convey the information, saying \"The galaxy is on Orion's belt.\" This cryptic message is in place to protect the information from being easily understood, even under duress.

Gemini : The Arquillian in the jeweler's head didn't simply tell Jay that the galaxy was on his cat's collar because of a combination of factors:
Limited communication: The Arquillian's physical form was weak and dying. It might not have had the capacity to speak clearly.
Symbolic language: The Arquillian may have used a metaphorical or symbolic language that Jay didn't understand at that moment.
Time constraint:The Arquillian was likely very weak and dying. It may not have had enough time to explain the location of the Galaxy clearly.
It's crucial to remember that the movie relies on a mix of science fiction and humor. While the Arquillian's cryptic message adds to the mystery, it also adds an element of humor and absurdity to the film's plot.

Llama-vid :The Arquillian in the jeweler's head was too occupied with his own situation to reveal the galaxy's location. When Jay first approaches him, the Arquillian is trying to escape from the Arquillians who are chasing him and his ship. He is focused on his own survival and does not realize Jay is also a Men in Black agent. When Jay flashes his neuralizer, the Arquillian realizes Jay is not an alien and is not a threat. At this point, the Arquillian is just trying to get away from the Arquillians and their ship. He does not want to reveal any information that could help the Arquillians find him.

Moviechat : In this video, we see a man in a grey shirt walking through a dimly lit building at night. As he walks, we see other people walking by as well. In the background, we can see the man in the blue shirt with a silver watch on his wrist. He is walking down the hallway as well. The man in the grey shirt is wearing a black shirt with a white collar as he goes down the stairs. There is a black and white photo of a person holding a camera that is displayed on the wall behind him. We see another man in the dark suit in the room with the grey shirted man, and a man in the black and white shirt in the room as well. Finally, we see a man in a black and white outfit looking up and down the hallway.

Large world model :The Arquillian in the jeweler's head simply told Jay that the galaxy was on his cat's collar as a playful and humorous interaction. It seems to be a lighthearted and amusing moment between the two. The Arquillian in the jeweler's head was likely a character or a character in the jeweler's head who was present during the interaction. By simply telling Jay that the galaxy was on his cat's collar, the Arquillian was adding a touch of humor and entertainment to their conversation.

Figure 24: Qualitative results of spoiler questions skill

**Linking multiple events:**

**System prompt :**
You play two roles: a human asking questions related to a video and an intelligent chatbot designed to help people find information from a given video.
Your task is to generate question-answer pairs specifically related to linking multiple events in the video content.
You will first play the role of a human who asks questions that link multiple events together in the video, and then play the role of an AI assistant that provides information based on the video content.
##TASK:
Users will provide information about the video, and you will generate a conversation-like question-and-answer pairs specifically focusing on linking multiple events together in the video to make the questions comprehensive across the video.
Generate TWENTY descriptive and conversational-style questions and their detailed answers based on the given information, specifically related to linking multiple events together in the video.
##INSTRUCTIONS:
- The questions must be conversational, as if a human is asking them, and should directly relate to linking multiple events together in the video.
- The answers must be detailed, descriptive, and should directly reference the information provided.
- The number of events to link together can vary from 2 to any number of events.
Please generate the response in the form of a list of Python dictionaries as strings with keys 'Q' for question and 'A' for answer. Each corresponding value should be the question-and-answer text respectively.
For example, your response should look like this: [{\Q\: \Your question here...\, \A\: \Your answer here...\},{\Q\: \Your question here...\, \A\: \Your answer here...\}].
Make sure to avoid to put double quotes inside string with double quotes, use single quotes instead. For example, use \I derived 'John's car\ yesterday\ instead of 'I derived \John's car\ yesterday' .
please only output the required format, do not include any additional information.
Remember well the output format of ONLY A PYTHON LIST as output and DON'T output the python shell because I will use python ast library to parse your output list.
## Few shot examples about the questions:
- What is the influence of event A on event B?
- How does event A lead to event B?
- What is the relationship between event A and event B?
- What is the impact of event A on event B?
- What is the connection between event A, event B, and event C?
**User prompt:**
The user input is {summary}.
Please generate the response in the form of a PYTHON LIST OF DICTIONARIES as strings with keys 'Q' for question and 'A' for answer. Each corresponding value should be the question-and-answer text respectively.
For example, your response should look like this: [{'Q': 'Your question here...', 'A': 'Your answer here...'},{'Q': 'Your question here...', 'A': 'Your answer here...'}].
DON'T output any other information because I will parse your output list.

Figure 25: Detailed prompt for Linking multiple events questions generation

**Character actions:**

**System prompt :**
You play two roles: a human asking questions related to a video and an intelligent chatbot designed to help people find information from a given video.
Your task is to generate a question-answer pairs specifically related to each character actions through the whole video content.
Your task is to first play the role of a human who asks questions about each character actions through the whole video content. and then play the role of an AI assistant that provides information based on the video content.
------
##TASK:
Users will provide information about a video, and you will generate a conversation-like question and answers pair specifically focusing on each character actions through the whole video content.
Generate one question for each character that summarize all the actions did through the whole video content.
------
##INSTRUCTIONS:
- The questions must be like a human conversation and directly related to each character actions through the whole video content.
- The answer must be detailed and descriptive that summarize all actions for each character in the video and should directly reference the information provided.
- Focus on both the visual and textual actions but focus more on the vision actions as these questions are designed for video understanding.
##SAMPLE QUESTIONS:
- {'Q1': 'What did ross do through this video?',  'A': 'At the beginning of the episode he drank coffee in central park , then went to his apartment then ate some pizza.'}
- {'Q1': 'Summarize all actions that chandler did in this video.', 'A': 'At the beginning of the episode he read a magazine then went to his work by taxi , and finally he went to Monica's apartment to set with his friends.'}
**User prompt:**
This is the episode summary: {caption}. \n
This is the episode script: {script}. \n
Please generate the response in the form of list of Python dictionaries string with keys 'Q' for question and 'A' for answer. Each corresponding value should be the question-and-answer text, respectively.
For the answer, please make it as a python list of actions in chronological order
For example, your response should look like this: [{'Q': 'Your question here...', 'A': ['Action 1','Action 2',...]},{'Q': 'Your question here', 'A': '['Action 1','Action 2',...]'}].
Please be very accurate and detailed in your response. Thank you!

Figure 26: Detailed prompt for sequence of character actions questions generation

**Temporal order of events:**

**System prompt:**
You play two roles: a human asking questions related to a video and an intelligent chatbot designed to help people find information from a given video.
##TASK:
Users will provide an episode Screenplay Script. Your task is to extract the events from this Screenplay Script. Ensure that the events are listed in chronological order
First read the Screenplay Script and think carefully to extract the all events.
------
##Few shot samples
Episode Screenplay Script: {user Screenplay Script}
Extract the events from this episode Screenplay Script:
The response should be in the format: ['Event A', 'Event B', 'Event C', 'Event D',...], ensuring that the event B is after event A and before Event C.
Remember well the output format of ONLY a PYTHON LIST of events and DON'T output the python shell because I will use python ast library to parse your output list.
**User prompt:**
Episode Screenplay Script: {script}
Extract the events from the Screenplay Script in a list
please provide the response in the format of PYTHON LIST of DON'T output any other information because I will parse your output list.
DON'T output any ' or ' in your response but use /u2019 for ' and /u2019s for 's and /u2019t for 't and s/u2019 for s' or s'

Figure 27: Detailed prompt for Temporal order of events questions generation

**Scene transitions:**

**System prompt:**
##TASK:
Users will provide an episode Screenplay Script. Your task is to extract scene transitions in from this script.
First read the Screenplay Script and think carefully to extract the transitions.
------
##Few shot samples
Episode Screenplay Script: {user Screenplay Script}
Extract the scene transitions from this episode Screenplay Script:
please provide the response in the format of PYTHON LIST of scene transitions like this example : ['scene A name', 'scene B name', 'scene C name',...], ensuring that the scene changed from A to B then C and so on.
Remember well the output format of ONLY a PYTHON LIST of events and DON'T output the python shell because I will use python ast library to parse your output list.
Scene names should be places name or location names where the scene is taking place such as home , cafe , bar , car and so on.
**User prompt:**
Episode Screenplay Script: {script}
Extract the scene transitions from this Screenplay Script in a list
please provide the response in the format of PYTHON LIST of scene transitions like this example : ['scene A name', 'scene B name', 'scene C name',...], ensuring that the scene changed from A to B then C and so on.
DON'T output any other information because I will parse your output list.

Figure 28: Detailed prompt for scene transitions questions generation

## Deep context understanding:

**System prompt:**
You play two roles: a human asking questions related to a video and an intelligent chatbot designed to help people find information from a given video.
##TASK:
Your task is to first play the role of a human who asks questions related to deep context understanding in the video and then play the role of an AI assistant that provides information based on the video content.
Users will provide human video summary and the video script,and you will generate a conversation-like question and answers pair specifically focusing on measuring the viewer's context understanding.
##INSTRUCTIONS:
- The questions must be conversational, as if a human is asking them, and should directly relate to deep context understanding for the video content.
- The answers must be detailed, descriptive, and should directly reference the information provided.
- The number of questions should be up to 20 questions and answers.
- The questions should be tricky and hard to answer to measure the viewer's context understanding.
- The answers must be detailed, descriptive, and should directly reference the information provided.
- It will be good if most of the questions are related to the visual content of the video.
-Again, the questions should be very tricky and hard to answer to measure the viewer's context understanding.
Please generate the response in the form of a list of Python dictionaries as strings with keys 'Q' for question and 'A' for answer. Each corresponding value should be the question-and-answer text respectively.
For example, your response should look like this: [{'Q': 'Your question here...', 'A': 'Your answer here...'},{'Q': 'Your question here...', 'A': 'Your answer here...'}].
please only output the required format, do not include any additional information.
If you want to type 's or 't and so on, please use \u2019s for 's and \u2019t for 't and so on.
Test your output by using the python ast library to parse your output list.
Remember well the output format of ONLY a PYTHON LIST as output
**User prompt:**
video summary: {caption}.
video transcript: {script}.
Please generate up to 20 questions and their answers in the form of list of Python dictionaries string with keys 'Q' for question and 'A' for answer. Each corresponding value should be the question-and-answer text respectively.
For example, your response should look like this: [{'Q': 'Your question here...', 'A': 'Your answer here...'},{'Q': 'Your question here...', 'A': 'Your answer here...'}].

Figure 29: Detailed prompt for deep context understanding questions generation

