# OpenReview forum: "InfiniBench: A Comprehensive Benchmark for Large Multimodal Models in Very Long Video Understanding"
_ICLR.cc/2025/Conference — Submitted to ICLR 2025_

### Official Review · Reviewer_dcjq · 2024-10-27

**Soundness:** 4
**Presentation:** 4
**Contribution:** 4
**Rating:** 8
**Confidence:** 5

**Summary:**

The paper presents a benchmark, called InfiniBench, for the evaluation of long video understanding. The dataset consists of 1219 videos. The average length of the videos is 52.59 minutes. There are 108.2K (video, question) pairs. The questions are divided into 9 categories. Some categories require the ability to make associations across a longtime span. Some categories require in-depth understanding and reasoning capabilities. It is a very interesting new benchmark for long video understanding.

**Strengths:**

The videos are very long with an average length 52 minutes.

The number of (question, answer) pairs is large (108k)

Some of the questions are unique such as spoiler questions, global appearance, and scene transitions.

Compared to the existing benchmarks, this benchmark contains much longer videos and contains some new interesting types of questions. It'll be very useful to the researchers who work on long video understanding.

**Weaknesses:**

The variety of TV show sources is limited since there are only 6 different TV shows.

**Questions:**

Can the authors comment on the limited variety of TV shows? What about sports events like NBA, NFL, Tennis, etc.

On GPT-4o evaluation, only 250 frames are selected. Are the 250 frames selected uniformly? Have you tried to reduce the frame size and squeeze more frames into GPT-4o?

Will all the videos be released to public? Are there any legal issues?

Are there text scripts (screenplay) associated with all the videos (movies and TV shows)?

---

> ### Author Response · Authors · 2024-11-25
> **Response 1, Part 1**
>
> ---
> > Q1: Can the authors comment on the limited variety of TV shows? What about sports events like NBA, NFL, Tennis, etc.
>
> We appreciate the reviewer's suggestion and agree that including a wider variety of video sources, such as sports events, could add value to future benchmarks. However, we argue that movies and TV shows are highly suitable and effective for assessing long-video understanding for the following reasons:
>
> 1. Diverse and Complex Contexts:
>
> * Movies and TV shows are rich in content, featuring intricate character relationships, evolving storylines, and multi-layered themes. These elements introduce dynamic and complex reasoning challenges beyond the repetitive or monotonic scenarios often found in daily-life videos like vlogs or live streams.
>
> * For instance, movies often include unexpected actions, rapid shifts in context, and non-linear narratives that demand a higher level of understanding, making them ideal for testing models' ability to handle long-term dependencies and reasoning tasks.
>
> 2. Variety of Skills Tested:
>
> * Our benchmark is designed to include diverse questions that evaluate different levels of video understanding, from surface-level observations to deeper reasoning about characters, events, and causality.
>
> * This holistic design ensures that the benchmark challenges models on a wide range of skills, as demonstrated by the low performance of current state-of-the-art models.
>
> 3. Limitations of Casual Videos:
>
> * Vlogs and daily-life videos often revolve around singular, straightforward activities (e.g., riding a bicycle or walking through a park). These scenarios typically lack the nuanced interplay between characters and the layered storytelling in cinematic content.
>
> * While casual videos may help test immediate perception tasks, they are less suited for evaluating the reasoning and complex relational understanding required for true long-video comprehension.
>
> 4. Storytelling Patterns:
>
> * While movies and TV shows follow storytelling patterns, these patterns are not uniform and vary greatly across genres, cultures, and creators. This inherent variability further enriches the benchmark by introducing a broad spectrum of reasoning and comprehension challenges.
>
> **Acknowledgment and Future Work:**
>
> We acknowledge that including additional categories of videos, such as sports events, could further diversify the benchmark and make it more comprehensive. We plan to explore incorporating these sources in an extended version of the work or future studies.
>
>
> ---
> > Q2: Will all the videos be released to public? Are there any legal issues?
>
>
> We appreciate the reviewer’s concern about copyright implications and ethical considerations related to using movies and TV shows in our dataset. Below, we clarify our approach and steps to ensure compliance with legal and ethical standards:
>
> 1. Use of Existing Datasets (e.g., MovieNet and TVQA):
>
> * We rely on publicly available datasets like MovieNet and TVQA, which provide downsampled versions of video content.
>
> * Both datasets are widely used in academic research and distribute videos in a legally compliant manner by heavily reducing the frame rate (e.g., 1 FPS). This downsampling transforms the videos into low-resolution derivatives primarily for research purposes, which may mitigate copyright concerns.
>
> * While we cannot definitively state that downsampling resolves all legal and ethical issues, its widespread use in academia suggests it is generally accepted within the research community.
>
> 2. Benchmark Without Redistribution of Videos:
>
> * Our benchmark does not redistribute video content directly. Instead, we provide annotations, question-answer pairs, and evaluation scripts.
> Users are required to independently download the videos from official sources (e.g., MovieNet or TVQA).
>
> * This ensures we do not claim ownership of the original video content, nor do we host or distribute it ourselves.
>
> * This approach aligns with practices used in existing benchmarks, where users are required to obtain the original images independently.
>
> 3. Ethical Considerations:
>
> * We acknowledge that relying on copyrighted material, even in downsampled form, can raise ethical questions.
>
> * We are committed to transparency and ensuring our benchmark is used responsibly. To that end, we will include a clear disclaimer stating:
>
> 1. The benchmark does not redistribute or modify original video content.
>
> 2. Users must adhere to the terms and conditions of the original video sources.
>
> **Conclusion:**
>
> While we cannot claim a definitive resolution of all legal or ethical concerns, our approach of using downsampled versions from widely accepted datasets and ensuring users obtain the videos independently reduces our direct responsibility for copyright compliance.
> If further legal concerns arise, we will explore ways to refine our approach in collaboration with legal experts and ensure the benchmark remains compliant and ethical.

---

> ### Author Response · Authors · 2024-11-25
> **Response 1, Part 2**
>
> ---
> > Q3:
>
> > Q3.1: On GPT-4o evaluation, only 250 frames are selected. Are the 250 frames selected uniformly?
>
> Yes, we sample 250 frames uniformly.
>
> > Q3.2: Have you tried to reduce the frame size and squeeze more frames into GPT-4o?
>
> 1. Influence of Input Frame Rate:
>
> * Feeding more frames intuitively improves accuracy, but the degree of improvement varies across models.
>
> * For instance, GPT-4o benefits the most from higher frame rates, while LLaVA-OV’s performance remains almost unchanged despite using an 8x higher frame rate.
>
> 2. Analysis of LLaVA-OV’s Behavior:
>
> * The limited benefit of higher frame rates for LLaVA-OV may be attributed to its training strategy.
>
> * LLaVA-OV is trained jointly on single images, multi-images, and videos.
>
> * This strategy employs a balanced visual representation approach, aggressively downsampling video inputs to ensure parity with image-based scenarios.
>
> * While effective for general tasks, this aggressive downsampling likely hurts LLaVA-OV’s ability to understand long videos, limiting its benefit from higher frame rates.
>
> 3. Skill-Specific Insights:
>
> * Specific skills benefit more from higher frame rates. For example, the ``local vision+text'' skill improves most as it relies on short sequential shots.
> * Increasing the frame rate reduces the chance of missing critical shots related to the answer, thereby boosting accuracy for such tasks.
>
> The results demonstrate that while higher frame rates generally improve performance, the degree of improvement depends on the model’s design and training strategy. Models like GPT-4o, optimized for sequential inputs, show significant gains, whereas models like LLaVA-OV, which aggressively downsample videos, see minimal benefits.
>
> | Qwen2VL            | Global Appearance | Scene transitions | character actions | Temporal questions | Local vision+text | Summarization | Deep context understanding | Spoiler questions | Linking Multiple Events | AVG acc | AVG score |
> |------|-------|-----|---------|--------|-----|-----|------|------|-----|----|-----|
> |                    | MCQ                     | MCQ               | MCQ               | MCQ                | MCQ               | Open-Ended    | Open-Ended                 | Open-Ended        | Open-Ended              |         |           |
> | Random Performance | 17                      | 17                | 16                | 42                 | 20                | N/A           | N/A                        | N/A               | N/A                     | N/A     | N/A       |
> | 250 Frames         | 36.6                    | 30.2              | 36.64             | 50.23              | 59.89             | 0.67          | 2.05                       | 1.39              | 2.82                    | 42.712  | 1.7325    |
> | 128 Frames         | 32.58                   | 28.64             | 34.59             | 49.33              | 54.98             | 0.59          | 1.87                       | 1.31              | 2.75                    | 40.024  | 1.63      |
> | 16 Frames          | 30.8                    | 20.83             | 32.2              | 46.59              | 42.9              | 0.3           | 1.53                       | 1.16              | 2.44                    | 34.664  | 1.3575    |
>
> | GPT-4o             | Global Appearance (ACC)  | Scene transitions | character actions | Temporal questions | Local vision+text | Summarization | Deep context understanding | Spoiler questions | Linking Multiple Events | AVG acc | AVG score |
> |--------------------|-------------------------|-------------------|-------------------|--------------------|-------------------|---------------|----------------------------|-------------------|-------------------------|---------|-----------|
> |                    | MCQ                     | MCQ               | MCQ               | MCQ                | MCQ               | Open-Ended    | Open-Ended                 | Open-Ended        | Open-Ended              |         |           |
> | Random Performance | 17                      | 17                | 16                | 42                 | 20                | N/A           | N/A                        | N/A               | N/A                     | N/A     | N/A       |
> | 250 Frames         | 45.98                   | 46.35             | 35.32             | 68.02              | 81.7              | 3.46          | 3.38                       | 2.72              | 3.47                    | 55.474  | 3.2575    |
> | 128 Frames         | 18.98                   | 29.84             | 17.92             | 43.12              | 22.1              | 1.78          | 0.37                       | 0.61              | 0.69                    | 26.392  | 0.8625    |
> | 16 Frames          | 20.37                   | 31.93             | 16.38             | 42.32              | 20.22             | 1.68          | 0.35                       | 0.63              | 0.65                    | 26.244  | 0.8275    |

---

> ### Author Response · Authors · 2024-11-25
> **Response 1, Part 3**
>
> ---
> > Q 3.2 (Continued)
>
> | InternVL             | Global Appearance (ACC)  | Scene transitions | character actions | Temporal questions | Local vision+text | Summarization | Deep context understanding | Spoiler questions | Linking Multiple Events | AVG acc | AVG score |
> |--------------------|-------------------------|-------------------|-------------------|--------------------|-------------------|---------------|----------------------------|-------------------|-------------------------|---------|-----------|
> |                    | MCQ                     | MCQ               | MCQ               | MCQ                | MCQ               | Open-Ended    | Open-Ended                 | Open-Ended        | Open-Ended              |         |           |
> | Random Performance | 17                      | 17                | 16                | 42                 | 20                | N/A           | N/A                        | N/A               | N/A                     | N/A     | N/A       |
> | 250 Frames         | N/A                     | N/A               | N/A               | N/A                | N/A               | N/A           | N/A                        | N/A               | N/A                     | N/A     | N/A       |
> | 128 Frames         | 25.89                   | 21.35             | 24.12             | 44.33              | 41.62             | 0.72          | 1.69                       | 1.27              | 2.53                    | 31.462  | 1.5525    |
> | 16 Frames          | 23.21                   | 20.83             | 25.18             | 44.82              | 31.95             | 0.7           | 1.54                       | 1.28              | 2.6                     | 29.198  | 1.53      |
>
>
> | Llava OV             | Global Appearance (ACC)  | Scene transitions | character actions | Temporal questions | Local vision+text | Summarization | Deep context understanding | Spoiler questions | Linking Multiple Events | AVG acc | AVG score |
> |--------------------|-------------------------|-------------------|-------------------|--------------------|-------------------|---------------|----------------------------|-------------------|-------------------------|---------|-----------|
> |                    | MCQ                     | MCQ               | MCQ               | MCQ                | MCQ               | Open-Ended    | Open-Ended                 | Open-Ended        | Open-Ended              |         |           |
> | Random Performance | 17                      | 17                | 16                | 42                 | 20                | N/A           | N/A                        | N/A               | N/A                     | N/A     | N/A       |
> | 250 Frames         | N/A                     | N/A               | N/A               | N/A                | N/A               | N/A           | N/A                        | N/A               | N/A                     | N/A     | N/A       |
> | 128 Frames         | 36.6                    | 23.95             | 25.911            | 45.49              | 48.6              | 0.55          | 1.79                       | 1.3               | 2.58                    | 36.1102 | 1.555     |
> | 16 Frames          | 41.51                   | 24.47             | 25.97             | 44.27              | 40.15             | 0.48          | 1.48                       | 1.33              | 2.3                     | 35.274  | 1.3975    |
>
>
> Using 250 frames per video is not applicable for LLava-OV and InternVL, as they are standard video models not designed to handle too-long videos.
> Accordingly, we hit the maximum context length for these models, which prevents us from running on higher sampling rates.
>
> ---
> > Q4: Are there text scripts (screenplay) associated with all the videos (movies and TV shows)?
>
> Yes, transcripts are available for all the movies and TV series included in our benchmark.
>
> * We collected these transcripts from publicly available sources on the internet and will publish them on our website to ensure convenience and facilitate future research.
>
> * However, as discussed earlier, we will not release the video content itself. Instead, we will refer users to the original sources, such as the MovieNet and TVQA datasets, where the videos can be independently accessed.
>
> This approach ensures compliance with copyright and ethical standards while making the benchmark resources easily accessible for the research community.

---

> > ### Comment · Reviewer_dcjq · 2024-11-26
> >
> > I appreciate the detailed responses and comments from the authors. The authors have addressed all my questions. I'd like to maintain my rating.
> >
> > A minor comment: in my statement "The variety of TV show sources is limited since there are only 6 different TV shows.", my concern was on the quantity (6) not the content. I agree that movies and TV shows in general are great for evaluating video understanding capabilities.

---

> > > ### Author Response · Authors · 2024-12-01
> > > **Thanks**
> > >
> > > We sincerely thank you for your thoughtful feedback and confirming that we have addressed your questions. We also greatly appreciate your detailed comments and the time you’ve taken to evaluate our work and for maintaining your positive rating.

---

### Official Review · Reviewer_iJs3 · 2024-11-01

**Soundness:** 3
**Presentation:** 2
**Contribution:** 2
**Rating:** 6
**Confidence:** 4

**Summary:**

In this work, the authors propose an InfiniBench for very long video understanding. To contain local/global events and understand visual/contextual content, they define a long video understanding covering nine skills through four critical aspects in movies and tv shows.

**Strengths:**

1 Important Topic. Long-form video understanding is a challenging but important problem. Hence, how to develop benchmark to evaluate this problem is critical.

2 Experiments. The experimental results are sufficient to support the claim of benchmark.

**Weaknesses:**

1 Similar work has been proposed in the literature. For example,  [MoVQA: A Benchmark of Versatile Question-Answering for Long-Form Movie Understanding, arXiv:2312.04817].  Please clarify the difference.

2 The writing and paper organization is not good. Please refine it for easy reading.

**Questions:**

Please see weakness.

---

> ### Author Response · Authors · 2024-11-25
> **Response 1**
>
> ---
> > **Q1: Compare with MoVQA.**
>
> We appreciate the contributions of MoVQA and acknowledge its value in advancing video question-answering research. However, our benchmark is significantly larger in scale and broader in scope, covering a wider range of skills, video durations, and models.
>
> **[Key Differentiation Points:]**
>
> 1. Scale:
>
>     * Our benchmark is 5× larger than MoVQA, with 108.2K QA pairs compared to 21.953K QA pairs in MoVQA.
>
> 2. QA Diversity:
>
>     * We support both multiple-choice questions (MCQ) and open-ended evaluations, while MoVQA is limited to MCQ only.
>
> 3. Video Sources:
>
>     * Our dataset features videos from both TV shows and movies, whereas MoVQA focuses solely on movies.
>
> 4. Video Length:
>
>     * The average video length in our benchmark is significantly longer (52.59 minutes) compared to MoVQA (16.53 minutes).
>
> 5. Model Coverage:
>
>     * We evaluate 10 long-video models, whereas MoVQA evaluates only 4 models (MPlug-Owl, Otter, VideoChatGPT, and VideoChat).
>
>
> **[Comparison with Other Recent Benchmarks:]**
>
> In addition to comparing with MoVQA, we have expanded our evaluation to include other recent benchmarks, such as Video-MME, LVBench, LongVideoBench, and MLVU.
>
> Here are some key highlights:
>
> 1. Video Length:
>
>     * Most benchmarks focus on short videos, with an average length of around 10 minutes.
>
>     * The exception is LVBench, which includes 1-hour-long videos, making it comparable in duration to our dataset.
>
> 2. Scale:
>
>     * Our benchmark is 70× larger than LVBench.
>
> 3. QA Types:
>
>     * We support both MCQ and open-ended QA, while other long-video benchmarks, such as LVBench and LongVideoBench, are limited to MCQ.
>
> 4. QA Resources:
>
>     * Our QA resources include the video script and summary, providing additional context for question-answering.
>
> 5. Challenging Capabilities:
>
>     * While most benchmarks, including LVBench and LongVideoBench, focus on visual understanding, our benchmark evaluates combined subtitle + visual understanding, making it more challenging.
>
> **Conclusion:**
>
> As shown in Table 1 of the revised paper, our benchmark surpasses MoVQA and other recent benchmarks in terms of scale, diversity, and the breadth of evaluation. We believe these advancements contribute significantly to the development of long-video understanding models.
>
> ---
> > **Q2: The writing and paper organization needs refinement.**
>
> Thank you for highlighting this issue. We have carefully revised and polished the writing and organization of the paper to ensure clarity and improve readability. Please refer to the revised version for the updated changes.

---

> ### Author Response · Authors · 2024-12-01
> **Kind reminder: We are looking forward to your reply**
>
> Dear Reviewer iJs3,
>
> We kindly ask if our response has addressed your concerns. Fortunately, we still have till December 3rd to discuss. Therefore, please feel free to share any additional questions or feedback, and we’ll be happy to provide further clarification.
>
> Best regards, The Authors

---

> > ### Author Response · Authors · 2024-12-03
> > **Kind reminder #2: We are looking forward to your reply**
> >
> > Dear Reviewer iJs3,
> >
> > We sincerely appreciate your dedicated time and effort in reviewing our paper.
> >
> > Since there are only a few hours remaining for reviewers to post messages to authors, we kindly ask if our additional clarifications and new results have sufficiently addressed your main concerns or if there are any remaining questions we can further address.
> >
> > Thank you once again for your valuable feedback. Incorporating these clarifications and experiments has helped strengthen the paper further.

---

### Official Review · Reviewer_oZkF · 2024-11-02

**Soundness:** 3
**Presentation:** 3
**Contribution:** 3
**Rating:** 8
**Confidence:** 4

**Summary:**

The paper proposes InfiniBench, a novel benchmark for long video understanding based on movies and TV shows. The benchmark has 108.2k question-answer pairs on 1,219 videos that average 52.59 minutes in length. The benchmark tests 9 different reasoning abilities including visual, long-context and local reasoning. This makes InfiniBench the largest-scale long video understanding benchmark to date. InfiniBench was constructed by combining and augmenting from two existing video benchmarks, TVQA and MovieNet. Most question types were generated by prompting GPT-4 with the transcript of the video while a custom pipeline was used to generate questions on changes in character appearance. The paper presents benchmark results of 8 long video understanding models, including 6 open source ones and 2 commercial ones, and discusses insights into their performance across various tasks.

**Strengths:**

* The presented benchmark has an impressive scale with 108.2k questions on 1,219 videos that average 52.59 minutes in length.
* There are 9 different question types that test long video understanding models across a variety of skills.
* The paper presents results of 8 long video models and draws interesting conclusions on their performance.
* There is a large gap between human performance and model performance, suggesting the benchmark has ample room for improvement.
* The paper has a good in-depth discussion of related work.

**Weaknesses:**

* The question-answer pairs in the benchmark were generated fully automatically without any human intervention. This raises questions about soundness and of the questions and potential bias. A human evaluation is performed on a subset of the data, but good human performance is no proof that questions are well-formed and free of hallucinations.
* Most questions are generated from transcripts that authors obtained online, but it is unclear what information these transcripts contain, whether they are complete and error-free. It is also unclear how much visual information the transcripts contain and therefore it is unclear to what degree this is a multimodal benchmark.
* The use of movies and TV shows raises questions about generalizability. Most MLLMs likely know the plots of popular movies and shows because their summaries or transcripts were part of their training data. So, they may be able to answer the questions in the dataset without any context, which is not the case for most videos from the web. The effect of this is not examined.
* It is unclear how much the benchmark relies on multimodal reasoning. Questions about movies and TV shows could often be answerable from subtitles alone, which are provided as context in the evaluation. It would be interesting to see an ablation that uses (1) No context, only the question itself (2) Only the question and subtitles (3) the question, subtitles and video frames.
* The copyright implications of using movies and TV shows and possibly releasing the dataset are not discussed and raise ethical concerns.
* Since the dataset has \~100 questions per video, it is likely that there are (near) duplicate questions. However there is no analysis of this and no mention of a filtering stage to remove duplicates.
* There are several issues with the presentation such as redundant figures, tables that are not referenced, and wrong references. The limitations section also exceeds the 10-page limit.

**Questions:**

Given the concerns listed above, I have doubts that this paper is suitable for publication at ICLR. I hope that authors can provide evidence to address my concerns as well as answers to the following questions.

* Could authors provide evidence of transcript quality? How accurate and complete are they? How much focus do they have on vision? Could authors provide examples?
* Why are multiple-choice questions evaluated by asking the model to generate an answer and then using GPT to match this answer to the options? Authors state in the appendix that the reason is that models often do not follow the prescribed answer format, but from my experience at least the larger VLMs are good at following instructions about the answer format.
* I am worried that using GPT for option matching introduces additional bias. I believe this could be measured by evaluating GPT or Gemini again by giving it the answer options in the prompt and asking it to respond with only the answer letter. Results could then be compared against the GPT-matched results.
  * Also to the above point, did authors verify that event ordering type questions get matched correctly with GPT? These answers only differ in their ordering of options, so I am wondering whether GPT matches them correctly.
* The benchmark was constructed using GPT, and GPT is the best performing model across all tasks. It would be interesting to quantify if there is bias towards GPT, e.g. by generating part of the data with Gemini and checking if relative model performance is consistent with the original benchmark.
* How are copyright concerns handled? Did authors obtain permission from the copyright owners to use the video material for this purpose and to reproduce this content in a publication? If the dataset will be publicly released, how are copyright concerns handled?
  * l. 198: “To address this limitation, we transformed the TVQA dataset from a collection of short clips into a long video dataset by gathering and sequencing the clips corresponding to each episode thereby reconstructing the full episode frames.“ How was this done and what data source was used?
* Appendix l. 12: “The remaining two skills, i.e., local visual questions and summarizing, do not need human verification, as the first one is adopted from the TVQA dataset, and the latter is scrapped from human responses on the web.” I do not fully agree with this statement since existing benchmarks and the humans writing the summaries that were pulled from the web could still contain errors. Do authors have evidence to the quality of TVQA annotations and summaries obtained from the web?
* How does the number of video frames provided affect the model accuracy?
* Appendix B is quite important to understand the evaluation results presented, so I think it would be better suited to be in the main text.
* Appendix B mentions that the benchmark videos have no audio, so video and subtitles are provided to the model separately. Does this mean that alignment between frames and subtitles is missing? Did authors measure the effect of this?
* Could authors explain how spoiler questions are generated and provide the prompt used?
* How does the “I don’t know” option affect results? How accurately does GPT match model answers to this option?
* Fig. 5 (left) is redundant with Tab. 3, so one of them should be removed.
* l. 363: The explanation of local vision and text questions is not clear. It is not explained what these questions are nor how they were generated.
* It would be good to have random accuracy in Tab. 5 for direct comparability. Then, Tab. 4 could be omitted.
* l. 482: “As shown in the table 5, MiniGPT4-video and LLaVA-NeXT-Interleave match lower than the random performance” What random performance is being compared to here? It would help to add this to the table as suggested above.
* l. 482, l. 505: How can a model’s performance be lower than random?
* l. 488: “One reason may be that eliminating the noisy information and focus on only the related information helps more in answering the questions“ How does the Goldfish model eliminate noisy information?
* For the human verification, how were human responses on open-ended questions evaluated?

Minor points

* Tab. 1: I would not agree with the “human” checkmark for InfiniBench since questions were generated fully automatically.
* Tab. 2 is never referenced.
* Appendix B: It would be helpful to express this in tabular form so readers can see at a glance how many frames and what modalities were used in each model.
* Tab. 5.: I would suggest to organize this into one big table with one column per task type. Also would be nice to visualize as a radar chart.
* It would be helpful to annotate question types in Sec 3.2.2 and Fig. 1 with whether they are MCQ or OE.
* It would be helpful to see a listing of modalities (vision, summary, transcript) used to generate each question.
* Please use \\citep for citations to place citations in parentheses.
* In tables, please right-justify numerical columns and use a consistent number of digits after the decimal point.
* Fig. 4: The font size in these charts is very small in print. I suggest increasing it. Also I would suggest to change the pie chart into a bar chart for easier readability.
* Fig. 5: Same concern as above about the font size.
* l. 373: Here, the reference to Fig. 4 is repeated, but Fig. 5 is wrongly referenced. Suggest correcting this sentence to refer to Fig. 3\.
* l. 406: Broken reference.
* l. 413: The reference should point to Sec. B in the supplementary material.

**Details Of Ethics Concerns:**

The following are my original concerns which have been mitigated:

I have a concern about potential copyright infringement in this work. The proposed dataset is based on copyrighted content (video frames and subtitles of movies and TV shows) that authors have downloaded and used for experiments. The paper also includes figures of frames from TV shows. It is unclear whether the authors obtained permission from copyright owners for their use of the data. Authors do not mention whether they intend to release the dataset publicly, but if they do, this would raise further concerns.

---

> ### Author Response · Authors · 2024-11-25
> **Response 1, Part 1**
>
> ---
> > Q1: A human evaluation is performed on a subset of the data, but good human performance is no proof that questions are well-formed and free of hallucinations.
>
> We appreciate the reviewer’s concern and would like to clarify the steps we have taken to ensure the quality and validity of our benchmark:
>
> [**Human Verification of 10% of the Data**]
>
> We conducted a human verification process on 10% of the dataset (10.8k questions) to assess the dataset's quality. By "dataset quality," we evaluate:
>
> By saying dataset quality we mean:
>
> 1. Validity of the Questions: Ensuring that the questions are relevant to the video.
>
> 2. Correctness of the Answers: Verifying whether the answers provided for valid questions are accurate.
>
> 3. Plausibility of Question-Answer Pairs: Checking if the format and phrasing of the pairs are clear and free of vagueness.
>
> This verification process ensures a holistic assessment of the benchmark’s reliability.
>
> **Results:**
>
> * On average, 95.8% of the questions were deemed correct and valid.
>
> * A breakdown of accuracy per skill is provided below:
>
> |Skill Name                  |Number of Questions| Accuracy |
> |----------------------------|-------------------|-------------------------|
> |Character Actions           |       667        |          94.9           |
> |Deep Context Understanding  |       2172        |          96.50           |
> |Global Appearance           |       135        |          89.62           |
> |Linking Multiple Events     |       2297        |          98.00           |
> |Scene Transitions           |       103        |          88.34           |
> |Spoiler Questions           |       43        |          95.34           |
> |Temporal Questions          |      2927        |          94.08           |
>
> We are continuously working to verify and correct the rest of the dataset and estimate that full verification will require approximately 4249 human hours.
>
> [**Human Verification of 100% of the Data**]
>
> The full dataset consists of:
>
> * 923 episodes, each requiring 3 hours on average for verification.
>
> * 296 movies, each taking approximately 5 hours to verify.
>
> To expedite this process, we have partnered with a data annotation company with ten full-time annotators working on the benchmark. Based on this setup, we estimate that the entire dataset will be fully verified and corrected within 20 working days.
>
> We believe these efforts demonstrate our commitment to ensuring the benchmark is free of hallucinations and robust enough for long-video understanding tasks.
>
> ---
> > Q2: **[Transcript Details]**
> Could authors provide evidence of transcript quality? How accurate and complete are they? How much focus do they have on vision? Could authors provide examples?
>
> 1. Role of Transcripts:
>
> * Transcripts are detailed documents created by movie or TV show writers. They provide comprehensive information beyond spoken dialogue, including:
> 1. Scene descriptions.
> 2. Context about settings, locations, and character actions.
> 3. Camera angles or shot compositions.
>
> * Transcripts serve as blueprints for visual and narrative elements, helping us extract visual insights and design challenging, reliable benchmark questions.
>
> 2. Transcript Example:
>
> ```
> Transcript of episode 1 season 1 of Friends TV shows:
> **[Scene: Central Perk, Chandler, Joey, Phoebe, and Monica are there.]**
> Monica: There's nothing to tell! He's just some guy I work with!
> Joey: C'mon, you're going out with the guy! There's gotta be something wrong with him!
> ...
> **(Ross gestures his consent.)**
> Joey: Strip joint! C'mon, you're single! Have some hormones!
> Ross: I don't want to be single, okay? I just... I just- I just wanna be married again!
> **(Rachel enters in a wet wedding dress and starts to search the room.)**
> ```
>
> As shown in the above example the transcript contains a lot of visual clues such as:
>
> * Rachel enters in a wet wedding dress and starts to search the room.
>
> * Scene: Central Perk, Chandler, Joey, Phoebe, and Monica are there.
>
> * Ross gestures his consent.
>
> 3. Subtitle Example:
>
> In contrast, the subtitle does not provide any visual clues therefore, we use it only during testing only.
>
> ```
> Subtitle of the same episode
> 1
> 00:00:54,012 --> 00:00:57,641
> There's nothing to tell.
> It's just some guy I work with.
>
> 2
> 00:00:57,892 --> 00:00:59,962
> You're going out with a guy.
> There's gotta be something wrong with him.
> ```
>
> By distinguishing between the roles of transcripts and subtitles, we ensure that the benchmark tests true long-video understanding during inference while using transcripts only for reliable benchmark construction.

---

> ### Author Response · Authors · 2024-11-25
> **Response 1, Part 2**
>
> ---
> > Q3: **[Data Leakage]**
> Most MLLMs likely know the plots of popular movies and shows because their summaries or transcripts were part of their training data. So, they may be able to answer the questions in the dataset without any context
>
> **[Blindness Experiment]**
>
> To genuinely assess the data leakage, we deliberately dropped the video and only fed the question and some context about the episode or the movie without any visual inputs.
>
> For instance, here is the input prompt in the blindness case:
>
> ``
> This is a question for a video from {show} {season_num} {episode_num}, use your knowledge to answer this question:
> {question}
> ''
>
> We have conducted the blindness experiments using two models, Qwen and GPT-4o.
> As shown in the tables below, in most skills, the blind models' performance is too close to the random performance.
>
> For instance, on the "global appearance" and the "scene transitions" skills, Qwen achieves 19.6 and 21, while GPT-4o achieves 20.8 and 22.5, approximately equal to the random performance of 17 for both skills.
>
>
> | Qwen | Global Apperance (ACC) | Scene transitions | character actions | Temporal questions | Local vision+text | Summarization | Deep context understanding | Spoiler questions | Linking Multiple Events | AVG acc | AVG score |
> | --- | --- | --- | --- | --- | --- | --- | --- | --- | --- | --- | --- |
> | Question Type | MCQ | MCQ | MCQ | MCQ | MCQ | Open-Ended | Open-Ended | Open-Ended | Open-Ended |  |  |
> | Random Performance     | 17                      | 17                | 16                | 42                 | 20                | N/A           | N/A                        | N/A               | N/A                     | N/A     | N/A       |
> | Video + sub + question | 36.6 | 30.2 | 36.64 | 50.23 | 59.89 | 0.67 | 2.05 | 1.39 | 2.82 | 42.712 | 1.7325 |
> | Question + Video Info | 19.64 | 21.35 | 35.05 | 46.57 | 39.71 | 0.28 | 1.7 | 1.48 | 2.6 | 32.464 | 1.515 |
> | Question | 18.75 | 19.27 | 29.62 | 45.49 | 38.29 | 0 | 0.97 | 0.76 | 1.7 | 30.284 | 0.8575 |
>
>
> | GPT-4o                 | Global Appearance (ACC)  | Scene transitions | character actions | Temporal questions | Local vision+text | Summarization | Deep context understanding | Spoiler questions | Linking Multiple Events | AVG acc | AVG score |
> |------------------------|-------------------------|-------------------|-------------------|--------------------|-------------------|---------------|----------------------------|-------------------|-------------------------|---------|-----------|
> |                        | MCQ                     | MCQ               | MCQ               | MCQ                | MCQ               | Open-Ended    | Open-Ended                 | Open-Ended        | Open-Ended              |         |           |
> | Random Performance     | 17                      | 17                | 16                | 42                 | 20                | N/A           | N/A                        | N/A               | N/A                     | N/A     | N/A       |
> | Video + sub + question | 45.98                   | 46.35             | 35.32             | 68.02              | 81.7              | 3.46          | 3.38                       | 2.72              | 3.47                    | 55.474  | 3.2575    |
> | Question + Video Info  | 20.83                   | 22.51             | 17.18             | 42.82              | 17.17             | 1.7           | 0.37                       | 0.68              | 0.7                     | 24.102  | 0.8625    |
> | Question               | 14.81                   | 24.08             | 15.78             | 42.35              | 16.44             | 1.75          | 0.36                       | 0.67              | 0.67                    | 22.692  | 0.8625    |
>
>
>
> **[Data Leakage vs. Common Sense]**
>
> In contrast, only the blind Qwen on one skill, the ``Character Actions'', achieves closer performance than the Qwen, which takes the visual input, 36.6 and 36, respectively.
> This could be interpreted as the model using its common sense to answer the question.
> The choices in this skill contain valid actions, and only their order is wrong.
> Thus, we argue that the model could perform well using common sense to order the events.
>
> To test our hypothesis, we assess the model performance on this skill as an open-ended question without choices.
> We leverage GPT-4o to score the models' outputs out of 5, where 0 is the worst and five is the best. The detailed prompt used while scoring is depicted in Figure 7.
> As expected, when we remove the visual input, the accuracy drops significantly from 0.79 to 0.003 as shown in the table below.
>
> |           Inputs           |    GPT-4o Score   |
> |----------------------------|-------------------|
> |         Questions          |      0.003        |
> |      Video + Questions     |       0.79        |

---

> ### Author Response · Authors · 2024-11-25
> **Response 1, Part 3**
>
> ---
> > Q4: **[Multi-Modal Benchmark]** It is unclear how much the benchmark relies on multimodal reasoning. It would be interesting to see an ablation that uses (1) No context, only the question itself (2) Only the question and subtitles (3) the question, subtitles and video frames.
>
> To evaluate the contribution of multimodal reasoning, we conducted experiments on Qwen using four input variants:
>
> 1. Video + Subtitle
> 2. Video
> 3. Question + Subtitle
> 4. Question
>
> **Findings:**
>
> 1. [Visual Clues]
>
>     * Dropping subtitles did not significantly impact performance, demonstrating that our benchmark focuses on visual cues and that textual information alone is insufficient to answer most questions.
>
> 2. [Blind Models]
>
>     * To assess the importance of visual inputs, we tested models using subtitles alone (termed Blind Models) without video frames.
>
>     * Results showed that blind models achieved near-random performance, emphasizing the critical role of visual inputs in answering questions.
>
> 3. [Blind and Deaf Models]
>
>     * Similarly, removing both video frames and subtitles (termed Blind and Deaf Models) resulted in random performance.
>
>     * This further highlights the significant role of visual inputs, with textual inputs alone contributing minimally, as evidenced by the similarity in performance between blind and blind-and-deaf models.
>
> Tables summarizing these findings have been included in the revised paper.
>
> | Qwen        | Global Apperance | Scene transitions | character actions | Temporal questions | Local vision+text | Summarization | Deep context understanding | Spoiler questions | Linking Multiple Events | AVG acc | AVG score |
> |----|-------|---|---|-----|------|------|-----|----|---|----|-----|
> |                     | MCQ                     | MCQ               | MCQ               | MCQ                | MCQ               | Open-Ended    | Open-Ended                 | Open-Ended        | Open-Ended              |         |           |
> | Random Performance  | 17                      | 17                | 16                | 42                 | 20                | N/A           | N/A                        | N/A               | N/A                     | N/A     | N/A       |
> | Video + Subtitle    | 36.6                    | 30.2              | 36.64             | 50.23              | 59.89             | 0.67          | 2.05                       | 1.39              | 2.82                    | 42.712  | 1.7325    |
> | Video               | 36.97                   | 28.12             | 36.97             | 49.06              | 47.56             | 0.35          | 1.69                       | 1.26              | 2.49                    | 39.736  | 1.4475    |
> | Question + Subtitle | 18.05                   | 25.65             | 17.85             | 44.48              | 20.79             | 1.86          | 0.43                       | 0.66              | 0.69                    | 25.364  | 0.91      |
> | Question            | 18.75   | 19.27 | 29.62 | 45.49 | 38.29 | 0 | 0.97 | 0.76 | 1.7 | 22.692  | 0.8625    |

---

> ### Author Response · Authors · 2024-11-25
> **Response 1, Part 4**
>
> ---
> > Q5: **[Copyrights]** The copyright implications of using movies and TV shows and possibly releasing the dataset are not discussed and raise ethical concerns.
>
> We appreciate the reviewer’s concern about copyright implications and ethical considerations related to using movies and TV shows in our dataset. Below, we clarify our approach and steps to ensure compliance with legal and ethical standards:
>
> 1. Use of Existing Datasets (e.g., MovieNet and TVQA):
>
>     * We rely on publicly available datasets like MovieNet and TVQA, which provide downsampled versions of video content.
>
>     * Both datasets are widely used in academic research and distribute videos in a legally compliant manner by heavily reducing the frame rate (e.g., 1 FPS). This downsampling transforms the videos into low-resolution derivatives primarily for research purposes, which may mitigate copyright concerns.
>
>     * While we cannot definitively state that downsampling resolves all legal and ethical issues, its widespread use in academia suggests it is generally accepted within the research community.
>
> 2. Benchmark Without Redistribution of Videos:
>
>     * Our benchmark does not redistribute video content directly. Instead, we provide annotations, question-answer pairs, and evaluation scripts.
> Users are required to independently download the videos from official sources (e.g., MovieNet or TVQA).
>
>     * This ensures we do not claim ownership of the original video content, nor do we host or distribute it ourselves.
>
>     * This approach aligns with practices used in existing benchmarks, where users are required to obtain the original images independently.
>
> 3. Ethical Considerations:
>
>     * We acknowledge that relying on copyrighted material, even in downsampled form, can raise ethical questions.
>
>     * We are committed to transparency and ensuring our benchmark is used responsibly. To that end, we will include a clear disclaimer stating:
>
> 1. The benchmark does not redistribute or modify original video content.
>
> 2. Users must adhere to the terms and conditions of the original video sources.
>
> **Conclusion:**
>
> While we cannot claim definitive resolution of all legal or ethical concerns, our approach of using downsampled versions from widely accepted datasets, combined with ensuring users obtain the videos independently, reduces our direct responsibility for copyright compliance.
> If further legal concerns arise, we will explore ways to refine our approach in collaboration with legal experts and ensure the benchmark remains compliant and ethical.
>
>
> ---
> > Q6: **[Duplicates]**
> Since the dataset has around 100 questions per video, it is likely that there are (near) duplicate questions.
>
> We appreciate the reviewer’s suggestion and agree that addressing potential duplicates is an important aspect of the benchmark creation process.
>
> **[Deduplication Methodology]**
>
> To ensure the dataset is free from (near) duplicate questions, we implemented the following approach:
>
> 1. Encoding and Similarity Calculation:
>
>     * We used M3-Embedding [1] to encode the questions and answer choices into vector representations.
>
>     * Cosine similarity was then calculated to identify potential duplicates.
>
> 2. Thresholds for Evaluation:
>
>     * To account for varying degrees of similarity, we evaluated three thresholds: 90%, 95%, and 98% cosine similarity.
>
> 3. Findings:
>
>     * As shown in Figure X of the revised paper, the vast majority of questions across all skills are unique, with no duplicates detected.
>
>     * Two skills, Temporal Questions and Character Actions, showed instances of potential duplicates.
>
> **[Analysis of Detected Duplicates]**
>
> Upon investigation, we found that the detected duplicates were false positives. For example, the following pair of questions was flagged as duplicates because they differ only by the words "before" and "after". However, this small difference completely changes the meaning of the question:
>
> 1. Q1: Did the event flashback to Phoebe completing a mile on a hippity-hop before turning thirty, happen before the event Monica makes breakfast with chocolate-chip pancakes?
>
> 2. Q2: Did the event flashback to Phoebe completing a mile on a hippity-hop before turning thirty, happen after the event Monica makes breakfast with chocolate-chip pancakes?
>
> These examples highlight the importance of semantic context in evaluating question similarity, as minor lexical differences can significantly alter meaning.
>
> **Conclusion**
>
> Based on our analysis, we are confident that the dataset is free from true duplicates, and our pipeline effectively identifies and handles potential near-duplicates. The false positives flagged by the similarity detection process underscore the complexity of semantic evaluation, especially in nuanced question construction.
>
>
> [1] Multi-Granularity, Multi-Linguality Multi-Functionality. "M3-Embedding: Multi-Linguality, Multi-Functionality, Multi-Granularity Text Embeddings Through Self-Knowledge Distillation."

---

> ### Author Response · Authors · 2024-11-25
> **Response1, Part 5**
>
> ---
> > Q7: Why are multiple-choice questions evaluated by GPT not exact matching?
>
> We appreciate the reviewer’s observation. The primary reason for using GPT-4o for evaluation instead of exact matching is to address cases where models do not strictly follow instructions and fail to output the selected choice directly.
>
> **Why GPT-4o Evaluation?**
>
> * The flexibility of GPT-4o allows us to focus on assessing the specific skill being tested rather than penalizing models for instruction-following errors.
>
> * This approach ensures that the evaluation emphasizes the core reasoning or understanding ability of the model, rather than its adherence to formatting requirements.
>
> **Comparison of Evaluation Methods**
>
> To assess the impact of these two evaluation strategies—exact matching (restrictive) and GPT-4o evaluation (flexible)—we conducted experiments on two models: GPT-4o and MiniGPT4_video.
>
> Findings:
>
> * GPT: Performs equally well under both exact matching and flexible evaluation.
> GPT strictly follows the instructions and outputs the exact choices, showing no advantage from flexibility.
>
> * MiniGPT4_video: Benefits significantly from the flexibility of GPT-4o evaluation.
> This model sometimes deviates from the instruction format but still demonstrates reasonable understanding of the question. Flexible evaluation allows us to capture its actual performance more effectively.
>
> **Conclusion and Future Reporting**
>
> * Inspired by this discussion, we will report results using both evaluation methods:
>
> 1. Exact Matching: A restrictive approach that directly evaluates instruction adherence.
>
> 2.GPT-4o Evaluation: A flexible approach that focuses on the skill being tested.
>
> This dual reporting will provide a more comprehensive understanding of model performance, balancing strict evaluation with flexibility where appropriate.
>
> |                              | Global Appearance | Scene transitions | Character actions | Temporal questions | Local vision+text  |
> |------------------------------|-------------------|-------------------|-------------------|--------------------|--------------------|
> | GPT4o (GPT matching)         | 45.98             | 46.35             | 35.3              | 68.02              | 81.70              |
> | GPT4o (Exact matching)       | 44.76             | 45.83             | 35.2              | 67.78              | 79.48              |
> | MiniGPT4-video (GPT)         | 3.57              | 1.5               | 2.7               | 39.54              | 13.97              |
> | MiniGPT4-video (Exact match) | 2.76              | 1.04              | 1.6               | 7.63               | 0.1                |
>
> This ablation only for 20 % of the benchmark
>
> ---
> > Q8: How does the number of video frames provided affect the model accuracy?
>
> 1. Influence of Input Frame Rate:
>
> * Feeding more frames intuitively improves accuracy, but the degree of improvement varies across models.
>
> * For instance, GPT-4o benefits the most from higher frame rates, while LLaVA-OV’s performance remains almost unchanged despite using an 8x higher frame rate.
>
> 2. Analysis of LLaVA-OV’s Behavior:
>
> * The limited benefit of higher frame rates for LLaVA-OV may be attributed to its training strategy.
>
> * LLaVA-OV is trained jointly on single images, multi-images, and videos.
>
> * This strategy employs a balanced visual representation approach, aggressively downsampling video inputs to ensure parity with image-based scenarios.
>
> * While effective for general tasks, this aggressive downsampling likely hurts LLaVA-OV’s ability to understand long videos, limiting its benefit from higher frame rates.
>
> 3. Skill-Specific Insights:
>
> * Specific skills benefit more from higher frame rates. For example, the ``local vision+text'' skill improves most as it relies on short sequential shots.
> * Increasing the frame rate reduces the chance of missing critical shots related to the answer, thereby boosting accuracy for such tasks.
>
> The results demonstrate that while higher frame rates generally improve performance, the degree of improvement depends on the model’s design and training strategy. Models like GPT-4o, optimized for sequential inputs, show significant gains, whereas models like LLaVA-OV, which aggressively downsample videos, see minimal benefits.

---

> ### Author Response · Authors · 2024-11-25
> **Response1, Part 6**
>
> ---
> > Q8 (Continued)
>
> | Qwen2VL   | Global Appearance  | Scene transitions | character actions | Temporal questions | Local vision+text | Summarization | Deep context understanding | Spoiler questions | Linking Multiple Events | AVG acc | AVG score |
> |--|---|-----|----|----|----|----|---|---|----|----|----|
> |                    | MCQ                     | MCQ               | MCQ               | MCQ                | MCQ               | Open-Ended    | Open-Ended                 | Open-Ended        | Open-Ended              |         |           |
> | Random Performance | 17                      | 17                | 16                | 42                 | 20                | N/A           | N/A                        | N/A               | N/A                     | N/A     | N/A       |
> | 250 Frames         | 36.6                    | 30.2              | 36.64             | 50.23              | 59.89             | 0.67          | 2.05                       | 1.39              | 2.82                    | 42.712  | 1.7325    |
> | 128 Frames         | 32.58                   | 28.64             | 34.59             | 49.33              | 54.98             | 0.59          | 1.87                       | 1.31              | 2.75                    | 40.024  | 1.63      |
> | 16 Frames          | 30.8                    | 20.83             | 32.2              | 46.59              | 42.9              | 0.3           | 1.53                       | 1.16              | 2.44                    | 34.664  | 1.3575    |
>
>
> | GPT-4o             | Global Appearance | Scene transitions | character actions | Temporal questions | Local vision+text | Summarization | Deep context understanding | Spoiler questions | Linking Multiple Events | AVG acc | AVG score |
> |--|---|-----|----|----|----|----|---|---|----|----|----|
> |                    | MCQ                     | MCQ               | MCQ               | MCQ                | MCQ               | Open-Ended    | Open-Ended                 | Open-Ended        | Open-Ended              |         |           |
> | Random Performance | 17                      | 17                | 16                | 42                 | 20                | N/A           | N/A                        | N/A               | N/A                     | N/A     | N/A       |
> | 250 Frames         | 45.98                   | 46.35             | 35.32             | 68.02              | 81.7              | 3.46          | 3.38                       | 2.72              | 3.47                    | 55.474  | 3.2575    |
> | 128 Frames         | 18.98                   | 29.84             | 17.92             | 43.12              | 22.1              | 1.78          | 0.37                       | 0.61              | 0.69                    | 26.392  | 0.8625    |
> | 16 Frames          | 20.37                   | 31.93             | 16.38             | 42.32              | 20.22             | 1.68          | 0.35                       | 0.63              | 0.65                    | 26.244  | 0.8275    |
>
>
> | InternVL             | Global Appearance  | Scene transitions | character actions | Temporal questions | Local vision+text | Summarization | Deep context understanding | Spoiler questions | Linking Multiple Events | AVG acc | AVG score |
> |--|---|-----|----|----|----|----|---|---|----|----|----|
> |                    | MCQ                     | MCQ               | MCQ               | MCQ                | MCQ               | Open-Ended    | Open-Ended                 | Open-Ended        | Open-Ended              |         |           |
> | Random Performance | 17                      | 17                | 16                | 42                 | 20                | N/A           | N/A                        | N/A               | N/A                     | N/A     | N/A       |
> | 250 Frames         | N/A                     | N/A               | N/A               | N/A                | N/A               | N/A           | N/A                        | N/A               | N/A                     | N/A     | N/A       |
> | 128 Frames         | 25.89                   | 21.35             | 24.12             | 44.33              | 41.62             | 0.72          | 1.69                       | 1.27              | 2.53                    | 31.462  | 1.5525    |
> | 16 Frames          | 23.21                   | 20.83             | 25.18             | 44.82              | 31.95             | 0.7           | 1.54                       | 1.28              | 2.6                     | 29.198  | 1.53      |

---

> ### Author Response · Authors · 2024-11-25
> **Response 1, Part 7**
>
> ----
> > Q8 (Continued)
>
> | Llava OV             | Global Appearance  | Scene transitions | character actions | Temporal questions | Local vision+text | Summarization | Deep context understanding | Spoiler questions | Linking Multiple Events | AVG acc | AVG score |
> |--|---|-----|----|----|----|----|---|---|----|----|----|
> |                    | MCQ                     | MCQ               | MCQ               | MCQ                | MCQ               | Open-Ended    | Open-Ended                 | Open-Ended        | Open-Ended              |         |           |
> | Random Performance | 17                      | 17                | 16                | 42                 | 20                | N/A           | N/A                        | N/A               | N/A                     | N/A     | N/A       |
> | 250 Frames         | N/A                     | N/A               | N/A               | N/A                | N/A               | N/A           | N/A                        | N/A               | N/A                     | N/A     | N/A       |
> | 128 Frames         | 36.6                    | 23.95             | 25.911            | 45.49              | 48.6              | 0.55          | 1.79                       | 1.3               | 2.58                    | 36.1102 | 1.555     |
> | 16 Frames          | 41.51                   | 24.47             | 25.97             | 44.27              | 40.15             | 0.48          | 1.48                       | 1.33              | 2.3                     | 35.274  | 1.3975    |
>
> Using 250 frames per video is not applicable for LLava-OV and InternVL, as they are standard video models not designed to handle too-long videos.
> Accordingly, we hit the maximum context length for these models, which prevents us from running on higher sampling rates.
>
> ---
> > Q9: Could authors explain how spoiler questions are generated and provide the prompt used?
>
> The spoiler questions in our benchmark are not generated by GPT-4o or any other AI model. Instead, they are sourced from the IMDB website, where all questions and answers are created by humans.
>
> Process for Spoiler Questions:
>
> 1. We scraped spoiler-related questions from IMDB, which are manually written and answered by human contributors.
>
> 2. To ensure quality and completeness, we filtered out any unanswered questions, keeping only those with valid human-provided answers.
>
> 3. We verify 10% of them and the human verification accuracy is 95.34%
>
> This approach guarantees that the spoiler questions are realistic and reflect genuine human reasoning.
>
>
> ---
> > Q10: How can a model’s performance be lower than random?
>
> We appreciate the reviewer’s insightful question. While it might seem counterintuitive, there are valid reasons why a model's performance can fall below random chance in a multiple-choice question (MCQ) setting.
>
> Key Reasons:
>
> 1. Overfitting or Bias Towards Incorrect Patterns:
>
> * Some models may overfit to spurious correlations or patterns in the training data, leading to systematic biases in their predictions.
>
> * Instead of distributing predictions randomly across all answer choices, the model might consistently select incorrect answers due to these biases, resulting in performance worse than random.
>
> 2. Instruction-Following Issues:
>
> * In some cases, the model might fail to correctly follow the format of the question or ignore the instruction to choose from the given options.
>
> * This behavior can result in answers that are invalid or unrelated to the choices, further reducing accuracy.
>
> ---
> > Q11: For the human verification, how were human responses on open-ended questions evaluated?
>
> For human verification of open-ended questions, we asked annotators to evaluate the model's responses on a five-point scale based on their correctness level relative to the ground truth (GT), similar to the evaluation process used by GPT-4o.
>
> This approach ensures a consistent and fair evaluation of open-ended responses by aligning human judgments with the predefined GT criteria.
>
> ---
> > Q12: Do authors have evidence to the quality of TVQA annotations and summaries obtained from the web?
>
> 1. TVQA Annotations:
>
> * The quality of TVQA questions and answers is well-documented in the TVQA paper (Section 3.2, Table 8).
>
> * It explicitly mentions that “The negative answers in TVQA are written by human annotators. They are instructed to write false but relevant answers to make the negatives challenging.”
>
> * This demonstrates that the questions and options are carefully crafted and verified by humans, ensuring their quality and relevance.
>
> 2. IMDB Summaries:
>
> * According to IMDB’s contribution guidelines [source](https://help.imdb.com/article/contribution/titles/plots/G56STCKTK7ESG7CP#):
> Contributors must follow strict instructions when submitting plot summaries.
>
> * Each contribution is reviewed and approved by the IMDB team before publication.
>
> * This rigorous process ensures that IMDB summaries are reliable, accurate, and already human-verified.

---

> > ### Author Response · Authors · 2024-11-26
> > **Response 1, part 8**
> >
> > > Q13 It would be helpful to see a listing of modalities (vision, summary, transcript) used to generate each question.
> >
> > The table below shows the different modalities used to generate each type of questions.
> >
> > |Skill name | Input Modality|
> > |-----------|---------------|
> > |Character actions | Summary +Transcript |
> > |Deep context understanding | Summary +Transcript |
> > |Global appearance | Vision |
> > |Linking multiple events | Summary|
> > |Scene transitions | Transcript |
> > |Spoiler questions | web scraping |
> > |Temporal questions | Transcript |
> > |Local vision text questions | Adopted from TVQA|
> > |Summarization | web scraping |

---

> ### Author Response · Authors · 2024-12-01
> **Kind reminder: We are looking forward to your reply**
>
> Dear Reviewer oZkF,
>
> We kindly ask if our response has addressed your concerns. Fortunately, we still have till December 3rd to discuss. Therefore, please feel free to share any additional questions or feedback, and we’ll be happy to provide further clarification.
>
> Best regards, The Authors

---

> > ### Comment · Reviewer_oZkF · 2024-12-03
> >
> > I’d like to thank the authors for their very elaborate response and the many new experiments performed. Overall, the authors were able to address most of my concerns while also making significant improvements to the paper. I am still concerned about potential bias towards GPT4. While my 3rd question (bias in eval protocol) has been mitigated by the experiments for Q7, my concern about bias in the questions still stands. Nevertheless, since most of my concerns have been addressed, I am increasing my score to 8\. I am also withdrawing my call for an ethical review based on the authors’ response regarding potential copyright issues.
> >
> > Please find my detailed responses below:
> >
> > * **Q1**: Thank you for the clarification\! From reading the paper, I got the impression that the human evaluation simply asked human labelers to answer the questions. I see that significant care is being taken to ensure data quality. I would suggest filtering out those QAs that labelers marked as wrong.
> > * **Q2**: Thank you for the response. This clarifies my understanding of what a transcript is and mitigates my concern of insufficient visual data being present. I’d suggest adding this to the paper or supplementary material.
> > * **Q3**: Thank you for this additional analysis. This somewhat mitigates my concern about prior knowledge existing in models. Interestingly, the blind Qwen model does well on character actions and local vision+context while the blind GPT4o is barely better than random. For the sake of transparency, it would be good to add this blindness experiment to the supplementary material.
> > * **Q4**: Thank you for this interesting analysis\! This mitigates my concern about multimodality. It’s great to see that both vision and subtitles contribute positively to accuracy and there are complimentary effects when combining them. The insights per question type are also very valuable.
> > * **Q5**: Thank you for the detailed response\! The fact that this dataset only consists of annotations on top of other datasets and does not include frames from the TV shows alleviates my ethical concerns.
> > * **Q6**: I appreciate the authors performing an additional analysis and including the results in the paper. This somewhat alleviates my concern about duplication. I would call the example given a near-duplicate though since the same reasoning is needed to answer both question, except that the answer is flipped.
> > * **Q7**: Thank you for this additional analysis\! Since open-source VLMs are not as good at instruction following, especially with large contexts, I think this evaluation method makes sense. But reporting the exact matching metric is a good idea since it allows for a cheaper evaluation. Another thing worth trying would be, instead of penalizing a model if its answer does not parse, select its answer by random choice, which gives it a 1-in-N chance of answering correctly and is more fair. (I’m not asking this for the rebuttal, but just giving this as a suggestion on how the gap between the evaluation methods could be closed.)
> > * **Q8**: Thank you for the many additional experiments and insightful findings\!
> > * **Q9**: Thank you for the additional explanation.
> > * **Q10**: I’m still not sure this explains performance lower than random, since any learned biases would have to actively steer the model away from the correct answer in order for the score to be lower than random.
> > * **Q11**: Thank you for the clarification\!
> > * **Q12**: Thank you for this reassuring clarification\!

---

### Official Review · Reviewer_147D · 2024-11-03

**Soundness:** 2
**Presentation:** 2
**Contribution:** 1
**Rating:** 3
**Confidence:** 4

**Summary:**

This paper introduces InfiniBench, an innovative and comprehensive benchmark focused on evaluating large multimodal models' performance in understanding very long videos. InfiniBench is notable for its ultra-long video duration (averaging 52.59 minutes per video) and massive question-answer pairs (108.2K), covering nine different skills including multiple-choice and open-ended questions. These questions are designed to be both diverse and human-centric, with videos primarily sourced from movies and TV shows. Experimental results show that even leading AI models like GPT-4V and Gemini 1.5 Flash face significant challenges in long video understanding, achieving average accuracies of only 49.16% and 42.72%, with mean scores of 3.22 and 2.71 (out of 5) respectively. This indicates that while these models perform relatively well on local skills, they still have limitations in skills requiring global reasoning and deep contextual understanding, such as scene transitions and movie spoiler questions. Open-source models generally perform below random chance on multiple-choice questions, highlighting long-sequence global reasoning as a major challenge for existing models. Additionally, models relying on both video and text information perform poorly without caption input, emphasizing the importance of processing both visual and textual information for long video understanding. The introduction of InfiniBench aims to fill the gap in long video understanding benchmarks, drive the development of open-source large language models, and motivate multimodal large models toward more human-like long video understanding and reasoning capabilities, despite current limitations such as video source restrictions and script dependency.

**Strengths:**

1. InfiniBench provides a comprehensive evaluation of large multimodal models' capabilities in long video understanding through including the longest video duration and a large number of question-answer pairs, as well as designing diverse question types (multiple-choice and open-ended questions) covering nine different skills, thus thoroughly examining models' performance across multiple dimensions of long video understanding.

2. By evaluating various models including both commercial and open-source models, InfiniBench reveals the challenges and limitations of existing models in long video understanding, especially in tasks requiring deep contextual understanding and critical thinking. This in-depth assessment helps identify model deficiencies and provides clear directions for future research and model improvements.

3. InfiniBench's design not only tests models' technical capabilities but also drives models toward more human-like understanding and reasoning abilities. Through proposing human-centric questions, such as movie spoiler questions, it promotes model performance improvement in long video understanding tasks, which is significant for achieving more advanced AI applications and advancing the field of artificial intelligence.

**Weaknesses:**

1. The benchmark only uses movies and TV shows for testing, which is too limited. It should include more types of videos that show different parts of real life, like nature documentaries or home videos. The problem is that movies and TV shows follow certain storytelling patterns, so AI models might just learn these patterns instead of truly understanding the videos. They should add more casual videos like vlogs and livestreams to make the testing more realistic.

2. The benchmark needs written scripts to create its questions and answers. This is a big problem because most real-world videos don't come with scripts. Without scripts or captions, the benchmark can't test how well AI models understand regular videos that people actually watch and share online.

3. InfiniBench's testing does not cover current mainstream open-source models such as Qwen2VL, LLaVA-Onevision, and InternVL2. This makes it difficult to obtain a more comprehensive and in-depth comparison between open-source and closed-source models.

4. In Table 1, the benchmark comparison is insufficient, especially regarding some recent video benchmarks such as Video-MME and LongVideoBench. Additionally, the authors' definition of "very long" is problematic - MLVU and MovieChat have only a 3-minute gap, yet MLVU is defined as very long. This is not reasonable.

**Questions:**

1. How is GPT-4V's scoring aligned with human evaluation?
2. Why weren't the latest models tested, and why wasn't there comparison and discussion of the latest benchmarks?

---

> ### Author Response · Authors · 2024-11-25
> **Response 1, Part 1**
>
> ---
> > **Q1: The benchmark only uses movies and TV shows, which is too limited.**
> **They should add more casual videos like vlogs and livestreams to make the testing more realistic.**
>
> We appreciate the reviewer's suggestion and agree that including a wider variety of video sources, such as vlogs or live streams, could add value to future benchmarks. However, we argue that movies and TV shows are highly suitable and effective for assessing long-video understanding for the following reasons:
>
> 1. Diverse and Complex Contexts:
>
> * Movies and TV shows are rich in content, featuring intricate character relationships, evolving storylines, and multi-layered themes. These elements introduce dynamic and complex reasoning challenges beyond the repetitive or monotonic scenarios often found in daily-life videos like vlogs or live streams.
>
> * For instance, movies often include unexpected actions, rapid shifts in context, and non-linear narratives that demand a higher level of understanding, making them ideal for testing models' ability to handle long-term dependencies and reasoning tasks.
>
> 2. Variety of Skills Tested:
>
> * Our benchmark is designed to include diverse questions that evaluate different levels of video understanding, from surface-level observations to deeper reasoning about characters, events, and causality.
>
> * This holistic design ensures that the benchmark challenges models on a wide range of skills, as demonstrated by the low performance of current state-of-the-art models.
>
> 3. Limitations of Casual Videos:
>
> * Vlogs and daily-life videos often revolve around singular, straightforward activities (e.g., riding a bicycle or walking through a park). These scenarios typically lack the nuanced interplay between characters and the layered storytelling in cinematic content.
>
> * While casual videos may help test immediate perception tasks, they are less suited for evaluating the reasoning and complex relational understanding required for proper long-video comprehension.
>
> 4. Storytelling Patterns:
>
> * While movies and TV shows follow storytelling patterns, these patterns are not uniform and vary greatly across genres, cultures, and creators. This inherent variability further enriches the benchmark by introducing a broad spectrum of reasoning and comprehension challenges.
>
> **Acknowledgment and Future Work:**
>
> We acknowledge that including additional categories of videos, such as vlogs, live streams, or documentaries, could further diversify the benchmark and make it more comprehensive. We plan to explore incorporating these sources in an extended version of the work or future studies.
>
> ---
> > **Q2: Without scripts or captions, the benchmark can't test how well AI models understand regular videos people watch and share online.**
>
> We appreciate the reviewer’s observation and would like to clarify an important distinction regarding the use of transcripts and subtitles in our benchmark:
>
> **[Transcript vs. Subtitles]**
>
> 1. Role of Transcripts:
>
>     * Transcripts are detailed documents created by movie or TV show writers. They provide comprehensive information beyond spoken dialogue, including:
>         1. Scene descriptions.
>         2. Context about settings, locations, and character actions.
>         3. Camera angles or shot compositions.
>
>     * Transcripts serve as blueprints for visual and narrative elements, helping us extract visual insights and design challenging, reliable benchmark questions.
>
>     * Key Point: Transcripts are used only during the benchmark creation process to ensure robustness and question diversity, not during inference or evaluation.
>
> 2. Role of Subtitles:
>
>     * Subtitles focus solely on translating spoken dialogue into text, typically extracted by transcribing the video's audio.
>
>     * Subtitles are optional inputs for the AI model during inference, representing an additional modality when available.
>
> 3. Clarifying the Inputs:
>
>     * During inference, the model's input consists of video frames and, optionally, subtitles, as shown in Table 3.
>
> * Transcript Example:
>
> ```
> Transcript of episode 1, season 1 of Friends TV shows:
> [Scene: Central Perk, Chandler, Joey, Phoebe, and Monica are there.]
> Monica: There's nothing to tell! He's just some guy I work with!
> ...
> (Ross gestures his consent.)
> Joey: Strip joint! C'mon, you're single! Have some hormones!
> (Rachel enters in a wet wedding dress and starts to search the room.)
> ```
>
> * Subtitle Example:
>
> ```
> Subtitle of the same episode
> 1
> 00:00:54,012 --> 00:00:57,641
> There's nothing to tell.
> It's just some guy I work with.
>
> 2
> 00:00:57,892 --> 00:00:59,962
> There's gotta be something wrong with him.
> ```
> By distinguishing between the roles of transcripts and subtitles, we ensure that the benchmark tests true long-video understanding during inference while using transcripts only for reliable benchmark construction.

---

> ### Author Response · Authors · 2024-11-25
> **Response 1, Part 2**
>
> ---
> > **Q3: InfiniBench's testing does not cover current mainstream open-source models such as Qwen2VL, LLaVA-Onevision, and InternVL2**
>
> We have included three new recent long video models:
>
> 1. Qwen2VL
> 2. InternVL
> 3. LLava OV
>
> Due to time constraints, these experiments were conducted on 20% of the dataset.
>
> | Qwen2VL            | Global Appearance  | Scene transitions | character actions | Temporal questions | Local vision+text | Summarization | Deep context understanding | Spoiler questions | Linking Multiple Events | AVG acc | AVG score |
> |----|----|----|---|----|----|----|----|---|---|----|---|
> |                    | MCQ                     | MCQ               | MCQ               | MCQ                | MCQ               | Open-Ended    | Open-Ended                 | Open-Ended        | Open-Ended              |         |           |
> | Random Performance | 17                      | 17                | 16                | 42                 | 20                | N/A           | N/A                        | N/A               | N/A                     | N/A     | N/A       |
> | 250 Frames         | 36.6                    | 30.2              | 36.64             | 50.23              | 59.89             | 0.67          | 2.05                       | 1.39              | 2.82                    | 42.712  | 1.7325    |
> | 128 Frames         | 32.58                   | 28.64             | 34.59             | 49.33              | 54.98             | 0.59          | 1.87                       | 1.31              | 2.75                    | 40.024  | 1.63      |
> | 16 Frames          | 30.8                    | 20.83             | 32.2              | 46.59              | 42.9              | 0.3           | 1.53                       | 1.16              | 2.44                    | 34.664  | 1.3575    |
>
>
> | GPT-4o             | Global Appearance  | Scene transitions | character actions | Temporal questions | Local vision+text | Summarization | Deep context understanding | Spoiler questions | Linking Multiple Events | AVG acc | AVG score |
> |----|----|----|---|----|----|----|----|---|---|----|---|
> |                    | MCQ                     | MCQ               | MCQ               | MCQ                | MCQ               | Open-Ended    | Open-Ended                 | Open-Ended        | Open-Ended              |         |           |
> | Random Performance | 17                      | 17                | 16                | 42                 | 20                | N/A           | N/A                        | N/A               | N/A                     | N/A     | N/A       |
> | 250 Frames         | 45.98                   | 46.35             | 35.32             | 68.02              | 81.7              | 3.46          | 3.38                       | 2.72              | 3.47                    | 55.474  | 3.2575    |
> | 128 Frames         | 18.98                   | 29.84             | 17.92             | 43.12              | 22.1              | 1.78          | 0.37                       | 0.61              | 0.69                    | 26.392  | 0.8625    |
> | 16 Frames          | 20.37                   | 31.93             | 16.38             | 42.32              | 20.22             | 1.68          | 0.35                       | 0.63              | 0.65                    | 26.244  | 0.8275    |
>
>
> | InternVL             | Global Appearance | Scene transitions | character actions | Temporal questions | Local vision+text | Summarization | Deep context understanding | Spoiler questions | Linking Multiple Events | AVG acc | AVG score |
> |----|----|----|---|----|----|----|----|---|---|----|---|
> |                    | MCQ                     | MCQ               | MCQ               | MCQ                | MCQ               | Open-Ended    | Open-Ended                 | Open-Ended        | Open-Ended              |         |           |
> | Random Performance | 17                      | 17                | 16                | 42                 | 20                | N/A           | N/A                        | N/A               | N/A                     | N/A     | N/A       |
> | 250 Frames         | N/A                     | N/A               | N/A               | N/A                | N/A               | N/A           | N/A                        | N/A               | N/A                     | N/A     | N/A       |
> | 128 Frames         | 25.89                   | 21.35             | 24.12             | 44.33              | 41.62             | 0.72          | 1.69                       | 1.27              | 2.53                    | 31.462  | 1.5525    |
> | 16 Frames          | 23.21                   | 20.83             | 25.18             | 44.82              | 31.95             | 0.7           | 1.54                       | 1.28              | 2.6                     | 29.198  | 1.53      |

---

> ### Author Response · Authors · 2024-11-25
> **Response 1, Part 3**
>
> > Q3: (Continued)
>
> | Llava OV             | Global Appearance | Scene transitions | character actions | Temporal questions | Local vision+text | Summarization | Deep context understanding | Spoiler questions | Linking Multiple Events | AVG acc | AVG score |
> |----|----|----|---|----|----|----|----|---|---|----|---|
> |                    | MCQ                     | MCQ               | MCQ               | MCQ                | MCQ               | Open-Ended    | Open-Ended                 | Open-Ended        | Open-Ended              |         |           |
> | Random Performance | 17                      | 17                | 16                | 42                 | 20                | N/A           | N/A                        | N/A               | N/A                     | N/A     | N/A       |
> | 250 Frames         | N/A                     | N/A               | N/A               | N/A                | N/A               | N/A           | N/A                        | N/A               | N/A                     | N/A     | N/A       |
> | 128 Frames         | 36.6                    | 23.95             | 25.911            | 45.49              | 48.6              | 0.55          | 1.79                       | 1.3               | 2.58                    | 36.1102 | 1.555     |
> | 16 Frames          | 41.51                   | 24.47             | 25.97             | 44.27              | 40.15             | 0.48          | 1.48                       | 1.33              | 2.3                     | 35.274  | 1.3975    |
>
> After adding the new methods, a complete leaderboard can be seen in Table 5 of the paper.
>
> ---
> > Q4: In Table 1, the benchmark comparison is insufficient, especially regarding recent video benchmarks such as Video-MME and LongVideoBench. Additionally, the author's definition of "very long" is problematic - MLVU and MovieChat have only a 3-minute gap, yet MLVU is defined as very long. This is not reasonable.
>
> We appreciate the reviewer's feedback and agree that our previous taxonomy, based on a hard threshold (e.g., 10 minutes), is not robust.
>
> **[Updated Taxonomy]**
>
> Inspired by this discussion, we have adopted a more adaptive and reliable categorization method using K-means clustering:
>
> * We set $k=3$ to categorize benchmarks into three groups based on average video length.
>
> * This method adaptively determines the categories without relying on arbitrary thresholds.
>
> The updated categorization addresses inconsistencies in defining "very long" benchmarks and ensures a fair comparison.
>
> **[Expanded Comparison]**
>
> Additionally, we have included comparisons with recent benchmarks, such as Video-MME and LongVideoBench, in Table 1 of the revised paper.
>
> We believe these changes improve the reliability and comprehensiveness of our benchmark comparison. Thank you for raising this important point!
>
>
> ---
> > Q5: How is GPT-4V's scoring aligned with human evaluation?
>
> We conducted a human evaluation on 10% of the dataset to assess the alignment between GPT-4o's scoring system and human preferences.
>
> **Evaluation Methodology:**
>
> * We designed a simple GUI that displayed responses from two models side by side for each question.
>
> * Annotators were asked to select which model provided the better response based on quality and relevance.
>
> * We then measured the Pearson correlation between human preferences and GPT-4o's scoring or ranking systems.
>
> **Results:**
>
> * The correlation between human preferences and GPT-4o's scoring system was 96%, indicating that GPT-4o's scoring is highly reliable and closely aligned with human judgment.
>
> These results validate the robustness of GPT-4o's scoring system as a reliable evaluation metric.

---

> ### Author Response · Authors · 2024-12-01
> **Kind reminder: We are looking forward to your reply**
>
> Dear Reviewer 147D,
>
> We kindly ask if our response has addressed your concerns. Fortunately, we still have till December 3rd to discuss. Therefore, please feel free to share any additional questions or feedback, and we’ll be happy to provide further clarification.
>
> Best regards, The Authors

---

> > ### Author Response · Authors · 2024-12-03
> > **Kind reminder #2: We are looking forward to your reply**
> >
> > Dear Reviewer 147D,
> >
> > We sincerely appreciate your dedicated time and effort in reviewing our paper.
> >
> > Since there are only a few hours remaining for reviewers to post messages to authors, we kindly ask if our additional clarifications and new results have sufficiently addressed your main concerns or if there are any remaining questions we can further address.
> >
> > Thank you once again for your valuable feedback. Incorporating these clarifications and experiments has helped strengthen the paper further.

---

### Official Review · Reviewer_6qyh · 2024-11-03

**Soundness:** 3
**Presentation:** 3
**Contribution:** 2
**Rating:** 5
**Confidence:** 5

**Summary:**

This paper introduces InfiniBench, a video understanding benchmark dataset featuring the longest video duration (average 52.59 minutes per video) and the largest number of question-answer pairs (108.2K) to evaluate 9 different video understanding tasks.

The authors conducted comprehensive evaluations of existing large multimodal models (including commercial models like GPT-4V, Gemini 1.5 Flash, and open-source models). Experiments show that even leading AI models still face challenges in long video understanding, with the best models GPT-4V and Gemini 1.5 Flash achieving average accuracy rates of only 49.16% and 42.72% respectively.

**Strengths:**

1. The questions are comprehensive and well-structured, covering multiple dimensions and employing diverse construction strategies for different types of questions.
2. The evaluation methods are reasonable, adopting different assessment metrics for multiple-choice and open-ended questions.

**Weaknesses:**

1. The paper lacks discussion of related work. For example, benchmarks proposed in Video-MME, LVBench, and Long VideoBench published in June 2024 are very similar to InfiniBench.

2. Most of the question-answer pairs are generated by GPT-4o. Although multiple information sources were used as input, it's difficult to guarantee the quality of the dataset.

3. Part of the data comes from IMDB content, which likely appeared multiple times in the training corpus of LLMs used by video models, potentially leading to dataset leakage issues.

**Questions:**

1. Add references and discussions of related work.
2. It would be better to evaluate more long-video models (e.g., Qwen2VL) and different input frame rates (1, 8, 32, 128, and more).
3. Since most question-answer pairs are generated by GPT-4o, could this lead to inflated evaluation results for GPT-4o? Analysis is needed regarding dataset quality, hallucination rates, and potential information leakage issues.

---

> ### Author Response · Authors · 2024-11-25
> **Response 1, Part 1**
>
> ---
> > **Q1: The paper lacks discussion of related work, e.g., Video-MME, LVBench, and Long VideoBench.**
>
> We have expanded our evaluation to include other recent benchmarks, such as Video-MME, LVBench, and LongVideoBench.
>
> Here are some key highlights:
>
> 1. Video Length:
>
> * Most benchmarks focus on short videos, with an average length of around 10 minutes.
>
> * The exception is LVBench, which includes 1-hour-long videos, making it comparable in duration to our dataset.
>
> 2. Scale:
>
> * Our benchmark is 70× larger than LVBench.
>
> 3. QA Types:
>
> * We support both MCQ and open-ended QA, while other long-video benchmarks, such as LVBench and LongVideoBench, are limited to MCQ.
>
> 4. QA Resources:
>
> * Our QA resources include the video script and summary, providing additional context for question-answering.
>
> 5. Challenging Capabilities:
>
> * While most benchmarks, including LVBench and LongVideoBench, focus on visual understanding, our benchmark evaluates combined subtitle + visual understanding, making it more challenging.
>
>
> **Conclusion:**
>
> As shown in Table 1 of the revised paper, our benchmark surpasses the recent benchmarks in scale, diversity, and breadth of evaluation. We believe these advancements contribute significantly to the development of long-video understanding models.
>
> ---
> > **Q2: [The quality of the dataset] Most of the question-answer pairs are generated by GPT-4o. Although multiple information sources were used as input, it's difficult to guarantee the dataset's quality.**
>
> [**Human Verification of 10%**]
>
> We have conducted a human verification of our benchmark for 10% (10.8k questions) to verify the dataset's quality.
> The verification of the 10% of the data takes around 400 human hours.
> The results show the average of correct questions is (95.8), and humanly correct the rest of the dataset. So the final humanly verified set is 100% accurate. The remaining set is now considered as weak labels with 96% expected accuracy which we believe can be a valuable resource for training.  We are expanding the human verification process to cover the whole set which is expected to take around 4249 human hours.
>
> The detailed accuracy per skill is reported in the table below:
>
> |Skill Name                  |Number of Questions| Accuracy |
> |----|----|----|
> |Character Actions           |    667    |      94.9     |
> |Deep Context Understanding  |       2172        |     96.50     |
> |Global Appearance           |       135    |     89.62     |
> |Linking Multiple Events     |       2297        |    98.00   |
> |Scene Transitions           |       103        |          88.34     |
> |Spoiler Questions           |       43        |          95.34           |
> |Temporal Questions          |      2927        |          94.08           |
>
> [**Human Verification of 100%**]
>
> We have 923 episodes, whereas one requires 3 hours on average for verification. In addition, we have 296 movies, each taking around five hours to be verified.
> We have a contract with a data annotation company, with 10 annotators working full-time on our benchmark. Therefore, all the data should be verified and ready in 20 working days.
>
> ---
> > **Q3: [Dataset Leakage] Part of the data comes from IMDB content, which likely appeared multiple times in the training corpus of LLMs used by video models, potentially leading to dataset leakage issues.**
>
> **[Blindness Experiment]**
>
> To genuinely assess the data leakage, we deliberately drop the video and only feed the question and some context about the episode or the movie without any visual inputs.
>
> For instance, here is the input prompt in the blindness case:
>
> ``
> This is a question for a video from {show} {season_num} {episode_num}, use your knowledge to answer this question:
> {question}
> ''
>
> We have conducted the blindness experiments using two models, Qwen and GPT-4o.
> As shown in the tables below, in most skills, the blind models' performance is too close to the random performance.
>
> For instance, on the "global appearance" and the "scene transitions" skills, Qwen achieves 19.6 and 21, while GPT-4o achieves 20.8 and 22.5, approximately equal to the random performance of 17 for both skills.
>
> | Qwen | Global Apperance | Scene transitions | character actions | Temporal questions | Local vision+text | Summarization | Deep context understanding | Spoiler questions | Linking Multiple Events | AVG acc | AVG score |
> | --- | --- | --- | --- | --- | --- | --- | --- | --- | --- | --- | --- |
> | Question Type | MCQ | MCQ | MCQ | MCQ | MCQ | Open-Ended | Open-Ended | Open-Ended | Open-Ended |  |  |
> | Random Performance     | 17       | 17     | 16     | 42     | 20     | N/A     | N/A     | N/A     | N/A   | N/A     | N/A    |
> | Video + sub + question | 36.6 | 30.2 | 36.64 | 50.23 | 59.89 | 0.67 | 2.05 | 1.39 | 2.82 | 42.712 | 1.7325 |
> | Question + Video Info | 19.64 | 21.35 | 35.05 | 46.57 | 39.71 | 0.28 | 1.7 | 1.48 | 2.6 | 32.464 | 1.515 |
> | Question | 18.75 | 19.27 | 29.62 | 45.49 | 38.29 | 0 | 0.97 | 0.76 | 1.7 | 30.284 | 0.8575 |

---

> ### Author Response · Authors · 2024-11-25
> **Response 1, Part 2**
>
> ---
> Q3: (Continued)
>
> | GPT-4o                 | Global Appearance (ACC)  | Scene transitions | character actions | Temporal questions | Local vision+text | Summarization | Deep context understanding | Spoiler questions | Linking Multiple Events | AVG acc | AVG score |
> | --- | --- | --- | --- | --- | --- | --- | --- | --- | --- | --- | --- |
> |                        | MCQ                     | MCQ               | MCQ               | MCQ                | MCQ               | Open-Ended    | Open-Ended                 | Open-Ended        | Open-Ended              |         |           |
> | Random Performance     | 17                      | 17                | 16                | 42                 | 20                | N/A           | N/A                        | N/A               | N/A                     | N/A     | N/A       |
> | Video + sub + question | 45.98                   | 46.35             | 35.32             | 68.02              | 81.7              | 3.46          | 3.38                       | 2.72              | 3.47                    | 55.474  | 3.2575    |
> | Question + Video Info  | 20.83                   | 22.51             | 17.18             | 42.82              | 17.17             | 1.7           | 0.37                       | 0.68              | 0.7                     | 24.102  | 0.8625    |
> | Question               | 14.81                   | 24.08             | 15.78             | 42.35              | 16.44             | 1.75          | 0.36                       | 0.67              | 0.67                    | 22.692  | 0.8625    |
>
>
> **[Data Leakage vs. Common Sense]**
>
> In contrast, only the blind Qwen on one skill, the ``Character Actions'', achieves closer performance than the Qwen, which takes the visual input, 36.6 and 36, respectively.
> This could be interpreted as the model using its common sense to answer the question.
> The choices in this skill contain valid actions, and only their order is wrong.
> Thus, we argue that the model could perform well using common sense to order the events.
>
> To test our hypothesis, we assess the model performance on this skill as an open-ended question without choices.
> We leverage GPT-4o to score the models' outputs out of 5, where 0 is the worst and 5 is the best. The detailed prompt used while scoring is depicted in Figure 7.
> As expected, when we remove the visual input, the accuracy drops significantly from 0.79 to 0.003 as shown in the table below.
>
> |           Inputs           |    GPT-4o Score   |
> |----------------------------|-------------------|
> |         Questions          |      0.003        |
> |      Video + Questions     |       0.79        |
>
>
> ---
> > Q4. Evaluate more long-video models (e.g., Qwen2VL) and different input frame rates (1, 8, 32, 128, and more).
>
> We have included three new recent long video models:
>
> 1. Qwen2VL
> 2. InternVL
> 3. LLava OV
>
> In addition, we tested the best-performing model on our benchmark, GPT-4o, with different input frame rates. Due to time constraints, these experiments were conducted on 20% of the dataset.
>
> **[Findings]**
>
> 1. Influence of Input Frame Rate:
>
> * Feeding more frames intuitively improves accuracy, but the degree of improvement varies across models.
> * For instance, GPT-4o benefits the most from higher frame rates, while LLaVA-OV’s performance remains almost unchanged despite using an 8x higher frame rate.
>
> 2. Analysis of LLaVA-OV’s Behavior:
>
> * The limited benefit of higher frame rates for LLaVA-OV may be attributed to its training strategy.
> * LLaVA-OV is trained jointly on single images, multi-images, and videos.
> * This strategy employs a balanced visual representation approach, aggressively downsampling video inputs to ensure parity with image-based scenarios.
> * While effective for general tasks, this aggressive downsampling likely hurts LLaVA-OV’s ability to understand long videos, limiting its benefit from higher frame rates.
>
> 3. Skill-Specific Insights:
> * Specific skills benefit more from higher frame rates. For example, the ``local vision+text'' skill improves most as it relies on short sequential shots.
> * Increasing the frame rate reduces the chance of missing critical shots related to the answer, thereby boosting accuracy for such tasks.
>
> The results demonstrate that while higher frame rates generally improve performance, the degree of improvement depends on the model’s design and training strategy. Models like GPT-4o, optimized for sequential inputs, show significant gains, whereas models like LLaVA-OV, which aggressively downsample videos, see minimal benefits.

---

> ### Author Response · Authors · 2024-11-25
> **Response 1, Part 3**
>
> > Q4: (Continued)
>
> | Qwen2VL            | Global Appearance (ACC)  | Scene transitions | character actions | Temporal questions | Local vision+text | Summarization | Deep context understanding | Spoiler questions | Linking Multiple Events | AVG acc | AVG score |
> | --- | --- | --- | --- | --- | --- | --- | --- | --- | --- | --- | --- |
> |                    | MCQ                     | MCQ               | MCQ               | MCQ                | MCQ               | Open-Ended    | Open-Ended                 | Open-Ended        | Open-Ended              |         |           |
> | Random Performance | 17                      | 17                | 16                | 42                 | 20                | N/A           | N/A                        | N/A               | N/A                     | N/A     | N/A       |
> | 250 Frames         | 36.6                    | 30.2              | 36.64             | 50.23              | 59.89             | 0.67          | 2.05                       | 1.39              | 2.82                    | 42.712  | 1.7325    |
> | 128 Frames         | 32.58                   | 28.64             | 34.59             | 49.33              | 54.98             | 0.59          | 1.87                       | 1.31              | 2.75                    | 40.024  | 1.63      |
> | 16 Frames          | 30.8                    | 20.83             | 32.2              | 46.59              | 42.9              | 0.3           | 1.53                       | 1.16              | 2.44                    | 34.664  | 1.3575    |
>
>
> | GPT-4o             | Global Appearance (ACC)  | Scene transitions | character actions | Temporal questions | Local vision+text | Summarization | Deep context understanding | Spoiler questions | Linking Multiple Events | AVG acc | AVG score |
> | --- | --- | --- | --- | --- | --- | --- | --- | --- | --- | --- | --- |
> |                    | MCQ                     | MCQ               | MCQ               | MCQ                | MCQ               | Open-Ended    | Open-Ended                 | Open-Ended        | Open-Ended              |         |           |
> | Random Performance | 17                      | 17                | 16                | 42                 | 20                | N/A           | N/A                        | N/A               | N/A                     | N/A     | N/A       |
> | 250 Frames         | 45.98                   | 46.35             | 35.32             | 68.02              | 81.7              | 3.46          | 3.38                       | 2.72              | 3.47                    | 55.474  | 3.2575    |
> | 128 Frames         | 18.98                   | 29.84             | 17.92             | 43.12              | 22.1              | 1.78          | 0.37                       | 0.61              | 0.69                    | 26.392  | 0.8625    |
> | 16 Frames          | 20.37                   | 31.93             | 16.38             | 42.32              | 20.22             | 1.68          | 0.35                       | 0.63              | 0.65                    | 26.244  | 0.8275    |
>
>
> | InternVL             | Global Appearance (ACC)  | Scene transitions | character actions | Temporal questions | Local vision+text | Summarization | Deep context understanding | Spoiler questions | Linking Multiple Events | AVG acc | AVG score |
> | --- | --- | --- | --- | --- | --- | --- | --- | --- | --- | --- | --- |
> |                    | MCQ                     | MCQ               | MCQ               | MCQ                | MCQ               | Open-Ended    | Open-Ended                 | Open-Ended        | Open-Ended              |         |           |
> | Random Performance | 17                      | 17                | 16                | 42                 | 20                | N/A           | N/A                        | N/A               | N/A                     | N/A     | N/A       |
> | 250 Frames         | N/A                     | N/A               | N/A               | N/A                | N/A               | N/A           | N/A                        | N/A               | N/A                     | N/A     | N/A       |
> | 128 Frames         | 25.89                   | 21.35             | 24.12             | 44.33              | 41.62             | 0.72          | 1.69                       | 1.27              | 2.53                    | 31.462  | 1.5525    |
> | 16 Frames          | 23.21                   | 20.83             | 25.18             | 44.82              | 31.95             | 0.7           | 1.54                       | 1.28              | 2.6                     | 29.198  | 1.53      |

---

> ### Author Response · Authors · 2024-11-25
> **Response 1, Part 4**
>
> > Q4: (Continued)
>
> | Llava OV             | Global Appearance (ACC)  | Scene transitions | character actions | Temporal questions | Local vision+text | Summarization | Deep context understanding | Spoiler questions | Linking Multiple Events | AVG acc | AVG score |
> | --- | --- | --- | --- | --- | --- | --- | --- | --- | --- | --- | --- |
> |                    | MCQ                     | MCQ               | MCQ               | MCQ                | MCQ               | Open-Ended    | Open-Ended                 | Open-Ended        | Open-Ended              |         |           |
> | Random Performance | 17                      | 17                | 16                | 42                 | 20                | N/A           | N/A                        | N/A               | N/A                     | N/A     | N/A       |
> | 250 Frames         | N/A                     | N/A               | N/A               | N/A                | N/A               | N/A           | N/A                        | N/A               | N/A                     | N/A     | N/A       |
> | 128 Frames         | 36.6                    | 23.95             | 25.911            | 45.49              | 48.6              | 0.55          | 1.79                       | 1.3               | 2.58                    | 36.1102 | 1.555     |
> | 16 Frames          | 41.51                   | 24.47             | 25.97             | 44.27              | 40.15             | 0.48          | 1.48                       | 1.33              | 2.3                     | 35.274  | 1.3975    |
>
> Using 250 frames per video is not applicable for LLava-OV and InternVL, as they are standard video models not designed to handle too-long videos.
> Accordingly, we hit the maximum context length for these models, which prevents us from running on higher sampling rates.
>
>
> ---
> > Q5: Since most question-answer pairs are generated by GPT-4o, could this lead to inflated evaluation results for GPT-4o?
>
> We appreciate the reviewer’s suggestion and share the concern about ensuring a reliable and unbiased evaluation. To address this point, we conducted the following experiments:
>
> To address this point, we conducted several experiments:
> 1. **[Reliable Benchmark]**
> * We performed a human evaluation of the generated question-answer pairs.
> * The results show that the generated pairs align with human annotations by more than 95%, demonstrating that the benchmark is sufficiently reliable.
> 2. **[Poor Performance]**
> * Despite GPT-4o being used to generate the question-answer pairs, its evaluation performance is far from acceptable.
> * For example, it achieves only 35% accuracy on the "Character Actions" skill, while random performance is 16%. This indicates that the evaluation is not artificially inflated.
> 3. **[Input Richness]**
> * An important question arises: If GPT-4o can generate reliable, human-like question-answer pairs, why does it perform poorly on these questions?
> * The answer lies in the difference between the inputs used during data creation and evaluation:
> * During data creation, we provide GPT-4o with the transcript and the video summary, which contain rich visual and contextual information, including semantics, event context, character personalities, and other specifics.
> * To assess long-video understanding capabilities during the evaluation, we omit the transcript and the summary and provide only the video input.
> * This discrepancy in input richness justifies the gap between GPT-4o’s performance during testing and the benchmark creation.
>
> We hope these points clarify why the evaluation results are not inflated and further justify the reliability of the benchmark and experimental setup.

---

> ### Author Response · Authors · 2024-12-01
> **Kind reminder: We are looking forward to your reply**
>
> Dear Reviewer 6qyh,
>
> We kindly ask if our response has addressed your concerns. Fortunately, we still have till December 3rd to discuss. Therefore, please feel free to share any additional questions or feedback, and we’ll be happy to provide further clarification.
>
> Best regards, The Authors

---

> > ### Comment · Reviewer_6qyh · 2024-12-02
> > **Official Comment by Reviewer 6qyh**
> >
> > Thank you for the detailed responses and additional experiments. I am upgrading my rating to 5. Given the existence of several long-video benchmarks like Video-MME, LVBench, MLVU and LongVideoBench, while your benchmark provides valuable validation through human verification and blindness tests, I find that merely scaling up video length and dataset size represents an incremental rather than innovative advancement. The rigorous quality controls and comprehensive model evaluations are commendable, but future work would benefit from novel methodological contributions beyond quantitative expansion.

---

> > > ### Author Response · Authors · 2024-12-03
> > > **Thanks and Further Clarifications**
> > >
> > > We thank the reviewer for acknowledging our benchmark's usefulness and reliability and raising the score. We also appreciate your suggestion that future work could benefit from a stronger focus on novel methodological contributions beyond quantitative expansion.
> > > However, we would like to highlight several key aspects that underscore the significance and innovation of our work:
> > >
> > > 1. **Importance of Scale:**
> > >
> > >    * Existing long-video benchmarks are extremely limited in size; for example, the largest has only ~2k questions. In contrast, our benchmark is approximately 50× larger, containing 108k questions.
> > >    * Scale matters because smaller benchmarks may lead to misleading conclusions due to limited coverage. A larger dataset ensures robustness and diversity, enabling more reliable evaluation of long-video models.
> > > 2. **Introduction of New and Challenging Skills:**
> > >
> > >    * We introduce several novel and challenging evaluation skills, such as spoiler questions, global appearance reasoning, and linking events. These skills go beyond existing benchmarks and test nuanced multi-modal reasoning capabilities.
> > >
> > > 3. **Complexity of Movies and Series:**
> > >
> > >    * Movies and TV series present uniquely challenging scenarios with complex relationships, non-linear storytelling, and twists that demand deep multi-modal reasoning. Our benchmark leverages this complexity to push the boundaries of video understanding models.
> > >
> > > 4. **Holistic Coverage:**
> > >
> > >     * Compared to existing long-video benchmarks, ours is the most holistic in terms of the number of skills evaluated and the breadth of models covered. This makes it a comprehensive resource for the community and provides a clear roadmap for advancing long-video understanding research.
> > >
> > > 5. **Scalability Beyond Testing:**
> > >
> > >     * Due to its scale and diversity, our benchmark is not limited to evaluation; it can also serve as a valuable resource for training and pretraining video models, further accelerating progress in the field.
> > >
> > > 6. **Rigorous Quality Control:**
> > >
> > >     * Despite the large scale of the dataset, we have implemented rigorous human verification to ensure reliability. This careful balance between scale and quality sets our benchmark apart.
> > >
> > > We hope these points further highlight the contributions of our work and its potential impact on advancing long-video understanding. Thank you again for your valuable feedback and constructive suggestions, which have helped us strengthen our work.

---

### Author Response · Authors · 2024-12-01
**Summary of our Rebuttal**

We sincerely thank the reviewers and area chairs for their thoughtful feedback and constructive comments. Your insights have been instrumental in strengthening our work, improving its clarity, and providing additional evidence to support our claims.

In the revised manuscript, we have incorporated all feedback, with changes clearly highlighted in blue for your convenience.
Below, we summarize the key updates and experiments added to address your concerns:

1. Inclusion of More Models:

    * We have incorporated additional recent long-video understanding models into our benchmark to ensure they are comprehensive and holistic.
    * As a result, we have updated the findings in the benchmark, providing the community with more robust insights to guide future research directions.

2. Human Verification of the Benchmark:

    * We conducted a human verification of 10% of the benchmark and evaluated all models on this subset.
    * The strong correlation between model scores on the verified subset and the full benchmark demonstrates the reliability of our dataset.

3. Human Evaluation of Models:

    * We evaluated the correlation between our benchmark scores and human preferences, observing a high agreement of 95%.
    * This further validates the benchmark’s effectiveness in assessing model performance.

4. Dataset Leakage Analysis:

    * We performed additional experiments to assess dataset leakage and found it to be minimal due to the careful design of the benchmark.
    * This ensures the integrity of our evaluation and strengthens confidence in our results.

5. Influence of Subtitles:

    * We added ablation studies to analyze the role of subtitles, showing that their impact is limited.
    * This reinforces that our benchmark primarily evaluates visual understanding, focusing on vision-based reasoning rather than textual cues.

6. Enhanced Writing and Visuals:

    * We replaced nearly all figures with enhanced versions for better presentation and clarity.
    * Additionally, we revised several sections to improve readability and ensure a smoother flow of ideas.

We hope these updates comprehensively address your concerns and further demonstrate the robustness and value of our contributions. Thank you again for your invaluable feedback, which has greatly improved the quality of our work.

---

### Meta-Review · Area_Chair_zP6n · 2024-12-22

**Metareview:**

The paper presented a benchmark for long video understanding which is an important and challenging problem in the field. The paper received mixed ratings from the reviewers. There are some critical concerns raised by the reviewers. First, one reviewer is concerned about the diversity of the testing long videos. The videos are mainly from TV shows or movies which largely restricts the applicability of the benchmark to be used to model realistic video understanding problems. Another reviewer also mentioned that the testing videos are from only a limited number of channels which may bring bias in testing the performance of different video understanding models. Second, the reviewers are also criticizing the benchmark baseline models being used. There are several important video understanding models not tested in the benchmark. It brings limitations to understanding how challenging or useful the proposed benchmark is. Finally, there are also comments regarding the presentation quality of the submission. Based on these key points, AC decided to recommend a rejection for this time. The authors are encouraged to further polish the paper to submit it another time.

**Additional Comments On Reviewer Discussion:**

The reviewers requested further clarification on important details of the benchmark and asked for more experiments on state-of-the-art video understanding models. The reviewers are not fully satisfied with the authors' rebuttal.

---

### Decision · Program_Chairs · 2025-01-22

Reject